# Unsupervised detection of salt marsh platforms: a topographic method

Guillaume C. H. Goodwin[1], Simon M. Mudd[1], and Fiona J. Clubb[1,2]

[1]School of Geosciences, University of Edinburgh
[2]Institute of Earth and Environmental Science, University of Potsdam, Germany

*Correspondence to:* Guillaume C. H. Goodwin (g.c.h.goodwin@sms.ed.ac.uk)

**Abstract.**

Salt marshes filter pollutants, protect coastlines against storm surges, and sequester carbon, yet are under threat from sea level rise and anthropogenic modification. The sustained existence of the salt marsh ecosystem depends on the topographic evolution of marsh platforms. Quantifying marsh platform topography is vital for improving the management of these valuable landscapes. The determination of platform boundaries currently relies on supervised classification methods requiring near-infrared data to detect vegetation, or demands labour-intensive field surveys and digitisation. We propose a novel, unsupervised method to reproducibly isolate salt marsh scarps and platforms from a DEM, referred to as Topographic Identification of Platforms (TIP). Field observations and numerical models show that salt marshes mature into sub-horizontal platforms delineated by sub-vertical scarps. Based on this premise we identify scarps as lines of local maxima on a slope raster, then fill landmasses from the scarps upward, thus isolating mature marsh platforms. We test the TIP method using lidar-derived DEMs from six salt marshes in England with varying tidal ranges and geometries, for which topographic platforms were manually isolated from tidal flats. Agreement between manual and unsupervised classification exceeds 94% for DEM resolutions of 1 m, with all but one site maintaining an accuracy superior to 90% for resolutions up to 3 m. For resolutions of 1 m, platforms detected with the TIP method are comparable in surface area to digitised platforms, and have similar elevation distributions. We also find that our method allows for the accurate detection of local block failures as small as 3 times the DEM resolution. Detailed inspection reveals that although tidal creeks were digitised as part of the marsh platform, unsupervised classification categorizes them as part of the tidal flat, causing an increase in false negatives and overall platform perimeter. This suggests our method may benefit from combination with existing creek detection algorithms. Fallen blocks and high tidal flat portions, associated with potential pioneer zones, can also lead to differences between our method and supervised mapping. Although pioneer zones prove difficult to classify using a topographic method, we suggest that these transition areas should be considered when analysing erosion and accretion processes, particularly in the case of incipient marsh platforms. Ultimately, we have shown that unsupervised classification of marsh platforms from high-resolution topography is possible and sufficient to monitor and analyse topographic evolution.

# 1 Introduction

Salt marshes are highly dynamic ecosystems, sequestering on average 210 g $CO_2$ m$^{-2}$ yr$^{-1}$ through plant growth and decay (Chmura et al., 2003) and capturing additional inorganic sediment when they are submerged (Nardin and Edmonds, 2014). This productivity has allowed salt marshes to match historic sea level rise (Kirwan and Temmerman, 2009) and laterally expand when sediment inputs were sufficient (Kirwan et al., 2011). It also places them among the most valuable ecosystems in the world (Costanza et al., 1997), and they provide diverse ecosystem services such as flood attenuation (Möller and Spencer, 2002; Shepard et al., 2011), blue carbon sequestration (Chmura et al., 2003; Coverdale et al., 2014), and contaminant capture (Nelson and Zavaleta, 2012). Their economic value combined with their alarming retreat (Day et al., 2000; Duarte et al., 2008; Kirwan and Megonigal, 2013) makes monitoring the evolution of salt marshes crucial for developing management strategies that maintain the health of these ecosystems.

The most closely monitored properties of salt marsh ecosystems are ecological assemblages and elevation, as they are both essential to understand eco-geomorphic processes (Reed and Cahoon, 1992). For instance, elevation determines flooding frequency and therefore influences pioneer vegetation encroachment (Hu et al., 2015), which in turn affects vertical accretion through inorganic sediment capture (Pennings et al., 2005; Mudd et al., 2004, 2010). Individual plants also react to elevation by modifying their root to shoot length ratios, generating feedbacks between organic material build-up and sediment capture (Mudd et al., 2009). The variable intensity of these eco-geomorphic feedbacks enables salt marshes to accrete in response to variations in sea level, thus maintaining their place in the tidal frame (Kirwan and Temmerman, 2009; Crosby et al., 2016).

The objective detection and analysis of vegetation patterns is a mature field, with habitat mapping commonly undertaken through the analysis of spectral properties such as the Normalized Difference of Vegetation Index (NDVI) (Jucke van Beijma, 2015). NDVI mapping is now developed to the extent that it requires only a minimum of ground-truthing to determine the presence and type of vegetation (Hladik and Alber, 2014). This index has been shown to consistently differentiate vegetated areas from tidal flats (Tuxen et al., 2008) and flooded channels from dry land despite the sensitivity of classification algorithms (Belluco et al., 2006; Wang et al., 2007).

However, spectral data sources do not provide the topographic information necessary to fully understand morphodynamic processes: although Digital Elevation Models (DEMs) have been successfully generated from habitat maps in the Venice lagoon (Silvestri et al., 2003), additional influences on halophyte distribution such as groundwater circulation (Moffett et al., 2010, 2012) can lead to mismatches between topography and habitats (Hladik et al., 2013). These additional influences on habitat distribution prevent the reliable use of spectral data to infer topography. Furthermore, delineating salt marsh platforms exclusively from spectral sources encourages morphological studies to define salt marshes dominantly from an ecological perspective, whereas the physical setting, most notably the elevation within the tidal frame, plays a key role in maintaining ecosystem health (e.g., Morris et al., 2002).

The topographic data necessary to identify marsh platforms already exist: the proliferation of freely available high resolution topographic datasets from lidar or structure from motion (SfM) techniques means that DEMs with a grid cell size below 1 m are increasingly common on salt marshes, and offer vertical accuracies below 20 cm even without correcting for vegetation

(Sadro et al., 2007; Wang et al., 2009; Chassereau et al., 2011). At these resolutions, most scarps and channels are detectable on a DEM, and several automated topographic methods already allow the identification of tidal channel networks (Fagherazzi et al., 1999; Liu et al., 2015). However, contrary to spectral datasets, tools designed to accurately delineate the extent of salt marshes through means other than manual digitisation are lacking.

5      In this study, we propose an unsupervised method to topographically differentiate marsh platforms from tidal flats, which we refer to as Topographic Identification of Platforms (TIP). The TIP method aims to reproducibly and accurately delineate marsh platforms using only a DEM as input, while also reducing identification costs and enabling systematic topographic analyses of multiple salt marshes.

We here define salt marsh platforms as sub-horizontal surfaces in the coastal landscape, separated from surrounding intertidal 10  flats by steep scarp features. The processes that form salt marsh platforms can be described by ecological alternate stable states theory (Schroder et al., 2005) and geomorphic bifurcation models (Fagherazzi et al., 2006; Defina et al., 2007). These processes cause salt marshes to develop a distinctive, biologically-mediated topographic structure consisting of several sub-horizontal platforms, separated from tidal flats and from each other by a subvertical scarp and dissected by incising channels (Temmerman et al., 2007; Marani et al., 2007, 2013). The TIP method exploits this characteristic topography, which is clearly 15  visible on high-resolution DEMs and their associated slope rasters, to identify scarps and steep channel banks. As our method uses topographic signatures of marsh platforms, it will reflect the interplay between sedimentation, erosion, and biomass (Fagherazzi et al., 2012) rather than the distribution of specific macrophyte species. It should therefore be complementary to, rather than a replacement for, methods that detect plant zonation on marshes. We compare TIP-detected platforms with six manually digitised platforms from English marshes at varying grid cell sizes, demonstrating the potential of this method for 20  quantitative topographic analyses and short to mid-term monitoring.

## 2   Methodology

The TIP method automatically detects scarps and platforms of salt marsh systems from a DEM with no manual calibration requirements. Its general process is described in Fig. 1, and includes the possibility of filtering (step 1) and degrading (step 2) the DEM; the effects of both treatments are examined in the discussion. A slope raster is then generated by fitting a polynomial 25  surface to topographic data and taking the derivative of this surface (Hurst et al., 2012; Grieve et al., 2016) (step 3). Steps 4 and 5 are novel algorithms developed in this study to isolate scarps and platforms. The results of the isolation process are compared to manually generated platforms (step 6) to generate a comparison map (step 7).

### 2.1   Test sites

We test the TIP method on six sites in England, selected for the availability of airborne lidar data in the form of gridded 30  1 m resolution rasters, provided by the UK Environment Agency (http://environment.data.gov.uk/ds/survey/), and for the diversity of their morphologies and tidal ranges. Dataset metadata is available freely on the Environment agency website (https://data.gov.uk/dataset/lidar-composite-dtm-1m1). For each site, marsh platforms were digitised on an unfiltered and non-

degraded DEM at a scale of 1: 500, using the open-source software QGIS (step 6 in Fig. 1). Source data were flown in 2012 for all sites, unless noted otherwise. The locations of the selected sites are shown in Fig. 2.

Shell Bay, Dorset (S1) is a shallow bay with a spring tidal range of 2.4 m, located in Poole Harbour, a limited entrance bay (*sensu* Allen (2000)) protected from strong waves. The marshes in Shell Bay display jagged outlines, indicative of low wave
and tidal current stress (Leonardi and Fagherazzi, 2014). The Stour Estuary marshes (S2) 6 km upstream of the meso-tidal Stour mouth are subject to a spring tidal range of 3.8 m and fluvio-tidal currents due to their estuarine fringing position (*sensu* Allen (2000)), and therefore display more linear boundaries. The Stiffkey marshes (S3) are back-barrier marshes (Allen, 2000), which experience a 4.7 m spring tidal range and display signs of erosion and accretion. These recent perturbations to the marsh surface provide an interesting challenge for topographic detection of marsh extents. The macro-tidal Medway estuary marshes
(S4, spring tidal range of 6.4 m) were chosen due to the presence of numerous channels in the tidal flats. In order to test the ability of our method in regions with extreme tidal ranges, we also analysed two mega-tidal sites: Jenny Brown's Point marshes (S5, spring tidal range of 9.2 m) and the Parrett estuary (S6, spring tidal range of 11.8 m), where sand dunes, different elevations inside the tidal flats, fallen blocks and sunken platforms will test the limits of the method's ability to correctly delineate marshes in these environments.

## 2.2    Preprocessing Topographic Data

The TIP method isolates marsh platforms from a DEM up to their seaward limits by detecting the topographic signature generated by the development of salt marshes. The definition of landward boundaries can vary significantly with context, and may be defined by a vegetation zonation change (Mo et al., 2015), agricultural parcels, or infrastructure (Feagin et al., 2010). Topographic input data is therefore clipped to the landward limit of the platform, at the discretion of the user. In the preparation
stage, local slope is calculated from the DEM by fitting a second order polynomial surface (Hurst et al., 2012) with a window radius of three times the horizontal resolution of the DEM, selected because it is the minimum radius needed to calculate slope with this method. The DEM may be passed through a Wiener filter (Wiener, 1949; Robinson and Treitel, 1967) to reduce noise from lidar datasets and/or degraded by averaged subsampling before the determination of slope to match complementary datasets. The effect of enabling these optional treatments is further discussed in the results section. Although methods exist to
account for vegetation cover in the DEM (Hladik and Alber, 2012; Wang et al., 2009; Sadro et al., 2007; Chassereau et al., 2011; Montané and Torres, 2006), we chose not to apply these corrections as we wanted to ensure that the TIP method can be applied without information on the vegetation assemblages at a given site.

### 2.3    Scarp routing

Tidal flats and salt marshes occur mostly on low energy coasts (Allen, 2000), characterised by low local relief and slopes.
They therefore display similar local slope values, and this parameter alone is insufficient to differentiate between tidal flats and marsh platforms. Likewise, although marsh platforms are locally higher than tidal flats and channels, this may not be the case for complex depositional environments (e.g. marshes sheltered by a sand spit), where long-shore declivity may cause portions of the tidal flats to be higher than distant emergent platforms. Therefore, elevation alone, though it may be used to

visually identify salt marsh platforms, is insufficient for objective platform detection. We address this problem by investigating transition features such as channel banks and erosion scarps, which are outliers in both slope and elevation rasters. These features are commonly defined by steep local slopes, particularly in mature and eroding systems (Defina et al., 2007; Marani et al., 2013). Furthermore, scarps connect marsh platforms to tidal flats, and therefore represent a distinct break in elevation between the two. In this study, we focus on the identification of scarps and steep channel banks as a precursor to the detection of platforms, referred to as step 4 in Fig. 1.

To reduce computational costs, we delineate an initial search space to initiate the detection of scarps by isolating steep areas of the landscape, weighted by their elevation. We first calculate the relief of each pixel, $R_i$,

$$R_i = z_i - z_{min}, \tag{1}$$

where $z_i$ [dimensions L] is the elevation of the pixel and $z_{min}$ [L] is the minimum elevation in the DEM. We then divide this relief by the maximum relief in the DEM to get a dimensionless relief at each pixel, $R_i^*$:

$$R_i^* = \frac{R_i}{z_{max} - z_{min}} \tag{2}$$

A similar procedure is followed for slope, where $Rs$ [dimensionless] is determined by the slope at a pixel, $S_i$ minus the minimum slope $S_{min}$:

$$Rs_i = S_i - S_{min}, \tag{3}$$

and the dimensionless version is calculated as:

$$Rs_i^* = \frac{Rs_i}{S_{max} - S_{min}} \tag{4}$$

We then multiply these two metrics at each pixel to create the dimensionless parameter $P_i^*$ at each pixel:

$$P_i^* = R_i^* Rs_i^* \tag{5}$$

This dimensionless product is useful for highlighting steep areas at high elevations (Fig. 3): the higher the value of $P_i^*$, the steeper and higher the pixel is. $P_i^*$ could vary between 0 and 1, where a value of 0 would mean that a pixel was at both the lowest elevation and gradient in the DEM, and vice-versa for a value of 1.

We use the properties of the probability distribution function (pdf) of $P^*$ to define the first search space, which we call $Ss_1$. With the exception of macrotidal sites S5 and S6, the pdf of $P^*$ decreases monotonically with increasing $P^*$, and at sites S5 and S6 the pdf decreases monotonically after a peak value (Fig. 3a). When $f(P^*) < \max(f(P^*))$ and $P^* > \max(P^*)$, the derivative

of the pdf is negative and increasing, i.e., the slope of the pdf curve becomes gentler with increasing $P^*$. We therefore define the threshold value $P^*_{th}$ where the slope of the pdf is equal to a threshold slope, $Sp_{thresh}$, on the declining limb of the pdf curve (Fig. 3a). In this study we optimize the threshold value $Sp_{thresh}$ to improve the classification of each site, as described in the Results section. The first search space, $Ss_1$, is defined as those pixels where $P^* > P^*_{th}$, as shown in Fig. 3b. The search space $Ss_1$ is also schematically represented as grey cells in Fig. 4a (step 4.1)

We then define a square kernel $K_3$ of 3 cells in width around each cell in $Ss_1$. If more than one cell of $K_3$ is included in $Ss_1$, the cell containing the local slope maximum in $K_3$ is flagged as a first order scarp cell $Sc_1$. If one given $K_3$ already contains an $Sc_1$ cell that is not the central cell, the central cell will be flagged as an $Sc_1$ if and only if it is the next local maximum in $K_3$. This results in a patchwork of first order scarp cells (step 4.2 in Fig. 4a).

For each first order scarp cell $Sc_1$, we then flag two second order cells $Sc_2$ as neighbouring cells with the next steepest slopes contained in the search space and not in contact with each other (red outlines in Fig. 4b). If two $Sc_1$ cells are adjacent, only the cell with the higher slope will be flagged as a $Sc_2$ cell (step 4.3 in Fig. 4b). This generates a patchwork of first order cells (black outlines Fig. 4b) flanked by one or two second order cells (red outlines in Fig. 4b). Starting from the second order cells $Sc_2$, we prolong the scarps by finding the cell with the steepest slope that is not adjacent to another identified scarp cell of two lesser orders, within a $K_3$ kernel centred on the previously identified cell. For example, on the third iteration $Sc_3$ cells are identified in a $K_3$ kernel centred on a $Sc_2$ cell and must not be adjacent to an $Sc_1$ cell. Generally, $Sc_n$ cells are identified in a $K_3$ kernel centred on a $Sc_{n-1}$ cell and must not be adjacent to an $Sc_{n-2}$ cell. This routing procedure is applied in all kernels containing no more than two scarp cells and repeated until no cells fit the conditions or the order $n$ is equal to 100 (blue outlines, step 4.4 in Fig. 4b).

This procedure produces a large number of potentially misidentified scarps, as small creeks within the platform and in higher portions of the tidal flat tend to be selected during this procedure. We use a further algorithm to thin these scarps and eliminate creeks. The first procedure eliminates low elevation scarps. We first define a kernel of 9 cells in width $K_9$ (i.e., a square kernel of 81 pixels with the pixel being interrogated at its centre) and compare its maximum elevation $\max(ZK_9)$ to the 75th percentile $q_{75}$ of the entire DEM. Cells that do not satisfy the condition $\max(ZK_9) > ZK_{thresh} \times q_{75}$ are discarded from the finale ensemble of scarps (step 4.5 in Fig. 4c), where $ZK_{thresh}$ is a parameter which we optimize below. Each $K_9$ kernel containing less than 8 flagged cells is then discarded from the ensemble of scarps; after this procedure finishes we are left with the final ensemble of scarps (step 4.6 in Fig. 4d).

## 2.4 Platform identification

We identify marsh platforms based on the final ensemble of scarps (step 5 in Fig. 1). The final ensemble of scarps becomes a new search space $Ss_2$. We then create a square kernel 3 cells in width ($K_3$) around each cell in this new search space. Using this kernel we identify first order platform cells, $Pc_1$, which are defined as all cells within $K_3$ that have higher elevation values than the central cell of the kernel (i.e., those that are higher in elevation than the cells in the final scarp ensemble). We do this because platform cells are located at higher elevations than the scarp cells separating them from tidal flats. We use a kernel rather than a simple blanket elevation threshold over the entire DEM because longitudinal elevation variations may cause some

tidal flat cells to be higher than scarp cells. Each $Pc_1$ cell that is not adjacent to at least 2 other $Pc_1$ cells is considered a product of isolated situations and eliminated from the ensemble of platform cells.

Following this initial selection of platform cells, we proceed to iteratively fill the platforms. At this point, the initial ensemble of platform cells, $Pc_1$, is clustered around the final ensemble of scarps since we have only used a 3 pixel wide kernel centred on scarp cells to create the ensemble of $Pc_1$ cells. We then iterate using a filling algorithm. The first iteration uses the cells $Pc_1$, the second $Pc_2$, and so on. In each iteration of $Pc_n$ cells, new cells are identified using two kernels, one being larger than the other. First, we define a local elevation condition using an 11 pixel wide kernel $K_{11}$: we find the maximum elevation in this kernel and then subtract 20 cm to define the minimum local elevation for a platform pixel. The 20 cm leeway is applied to account for local elevation variations on the platforms. The algorithm will not identify as separate platforms separated by scarps less than this elevation threshold, so on microtial marshes this threshold can be lowered. We address this limitation in the discussion and appendix. The threshold is necessary to prevent the algorithm from excluding pools and slight depressions in the platform surface.

We then use a 3 pixel wide kernel $K_3$ within $K_{11}$ to identify any cells in the next iterations' platform ensemble ($Pc_{n+1}$). These cell must meet two conditions: i) that they are higher than the local elevation threshold identified with the 11 pixel kernel, and ii) that their distance to the nearest cell in the final scarp ensemble is greater than their distance to platform cells from previous iterations. The first condition is simply to ensure the platform is indeed a low relief surface, and the second is to ensure the iterative process fills the platform away from the scarps. The second condition is also necessary to ensure the platform filling process does not cross scarps. This iterative process is repeated until $n$ reaches an arbitrary value of 100, found to be sufficient to fill the entirety of the platform surface area for our sites.

This process results in platforms surfaces that are spatially continuous, but in some instances sections of the tidal flat with relatively high elevations may also have been identified as marsh platforms. These areas are lower than marsh platforms by the height of the scarp separating them. We filter these cells by using the elevation properties of the entire DEM. A number of authors have shown that there is a gap in the probability distribution of elevations in intertidal landscapes that separates the majority of tidal flats from the majority of marsh platforms in micro-tidal environments (e.g., Fagherazzi et al., 2006; Defina et al., 2007; Carniello et al., 2009). Such a separation, demonstrated by the decrease in probability between the grey and blue surfaces in Fig. 5, is also observed in our meso- and macro-tidal sites, including mega-tidal environments such as the Parrett estuary (Fig. 9). We search for this separation using the probability distribution of elevation, $pdf(z)$ of all cells $Pc_n$, divided in 100 elevations bins. We determine that the most frequent elevation bin $z_{max(pdf(z))}$ is the most likely to contain cells correctly assigned to the platform ensemble, as the relief of marsh platforms is lower than that of tidal flats. Therefore, only elevations lower than $z_{max(pdf(z))}$ may contain cells misidentified as marsh platforms.

We then must identify which cells from the population of cells lower than $z_{max(pdf(z))}$ form part of the platform, and which do not. To do this, we truncate low elevations that have a low probability (red curves in Fig. 5), to remove the long tail of low elevations from our initial platform identification. We take the probability distribution of the elevation of the remaining platform cells and calculate the mean probability $\bar{pdf}$ (i.e., we average the probability from the 100 bins). We then search for $rz_{thresh}$ consecutive elevation bins that lie below the elevation of the maximum probability elevation that have lower probabilities than

this average. The reason we use consecutive bins is that we do not want the minimum elevation to be determined by a single low probability elevation that has spuriously arisen from the binning process. Once we find $rz_{thresh}$ consecutive elevation bins meeting these criteria we remove all cells lower and including the highest cell that lies within the $rz_{thresh}$ consecutive bins. We optimize the parameter $rz_{thresh}$ below.

Having eliminated these low elevation, low probability cells, we also mark all cells higher than $z_{max(f(z))}$ as platform cells. This may still out leave pools and pans and platform edges remain jagged. Our final procedure aims to eliminate these artifacts using the following procedure: for a given value of the order $n$, we search in the ensemble of $Pc_n$ cells for cells that are surrounded by more than 6 $Pc$ cells of any order within a $K_3$ kernel. The 2 or less empty cells in $K_3$ are then attributed the order $n$-1. By iterating through values of $n$, starting with the order 100 and finishing with the order 2, we progressively fill pools and

jagged borders of the platform (Fig. 6a). Choosing 6 as the minimal number of platforms cells in each $K_3$ necessary to execute this "reverse filling" procedure, we ensure that no headlands are generated. We then integrate scarp cells that are connected to platform cells into the platform ensemble with an order greater than 100. We then repeat the "reverse filling" process (Fig. 6b) and execute low-elevation elimination procedure (See blue curves in Fig. 5) to obtain the final platform ensemble.

## 2.5   Performance metrics

In order to evaluate the performance of the TIP method, we compare its outputs to manually digitised platforms for all of our test sites (step 7 in Fig. 1). For each grid cell in the detected (automatically processed) and the reference (manually digitised) outputs, we assign the boolean value True to the marsh platform and False to the tidal flat. The results are classified as follows: true positives correspond to matching True cells in the tested and reference outputs, true negatives to matching False cells, false positives to True cells in the tested output that are False in the reference output, and false negatives to False cells in the tested

output that are True in the reference output. The performance of the method is then evaluated using three metrics based on the numbers of true positive (*TP*), true negative (*TN*), false positive (*FP*), and false negative (*FN*) cells respectively. The accuracy *Acc* (Fawcett, 2006) describes the likelihood of cells in the tested raster corresponding to the reference raster:

$$Acc = \frac{TP+TN}{TP+TN+FP+FN} \tag{6}$$

We also test the performance of the method by reporting two other metrics: the precision, *Pre*, and the sensitivity, *Sen*

(Fawcett, 2006). The precision represents the likelihood of the tested raster overestimating the positives compared to the reference:

$$Pre = \frac{TP}{TP+FP} \tag{7}$$

Conversely, the sensitivity *Sen*, represents the likelihood of the tested raster missing positives compared to the reference:

$$Sen = \frac{TP}{TP+FN} \tag{8}$$

If the results of the TIP method perfectly matched that of the manual digitisation, all three metrics would have a value of 1.

## 3 Results

### 3.1 Parameter optimisation

The TIP method contains three user-defined, non-dimensional parameters occurring in sequence during the detection process. The first parameter, $Sp_{thresh}$, determines the threshold value $P^*_{th}$ for the high-pass filter leading to the selection of the initial search space, shown in Fig. 3a. The parameter $Sp_{thresh}$ influences the solution of the equation $\frac{df}{dP^*} = Sp_{thresh}$. The second parameter, $ZK_{thresh}$ determines the condition on the refinement of existing scarps in the high-pass filter $max(ZK_9) > ZK_{thresh} \times q_{75}$, schematically represented in Fig. 4. The third parameter, $rz_{thresh}$ is used in the platform dispersion process to determine which percentage of the elevation range below $p\bar{d}f$ is maintained in the platform ensemble. In this study, these parameters were set to maximize the average accuracy $\bar{Acc}$ across test sites (Fig. 7): the optimized values ($Sp_{thresh}$=-2.0, $ZK_{thresh}$=0.85, $rz_{thresh}$=8) were used for the subsequent performance analysis. Users may modify these parameters as directed in the code documentation to better fit their study sites.

### 3.2 Validation and applicability

Figure 8 shows the performance of the TIP method for all six sites, discriminating between the use or absence of a Wiener filter and evaluating how the resolution of the topographic data influences the results. We also provide the full performance metrics in Appendix A (Tables A1 to A6). We find the method's accuracy to be on average 94.8% at the data's native resolution of 1 m, whether we apply a Wiener filter (Fig. 8a2) or not (Fig. 8a1). Degrading the DEM resolution still results in accuracy of above 90%, although it decreases to around 60% for microtidal site S1 at a resolution of 3 m. Applying a Wiener filter to the data causes a slight decrease in accuracy and precision (Fig. 8b2), but an increase in sensitivity (compare Fig. 8c2 to Fig. 8c1). Examining the results of all of the metrics shows that resolution degradation up to 3 m, well as the use of a Wiener filter, primarily causes an increase in false positives and therefore an overestimation in the extent of the marsh platform. For sites S2 to S6, we observe little change in performance metrics with resolution degradation up to 3 m.

We suggest that all three performance metrics should be used when optimising the TIP method for a study site, as no combination of two metrics provides comprehensive insight into TIP uncertainties. Furthermore, although average accuracies remain above 85% for resolutions of 4 to 5 m, we recommend caution when using the method at these resolutions, particularly in micro- to meso-tidal settings where features may be smoothed beyond the method's recognition capacities. Use of the TIP method is not recommended for resolutions coarser than 5 m due to the very low accuracies observed for our test sites, making this method adapted to high-resolution data sources such as airborne lidar or photogrammetry.

## 4 Discussion

### 4.1 Influence of site morphology on the TIP method

In order to examine the performance of the method in sites with varying morphological characteristics, we compare the probability distribution functions (pdf) of elevation from the digitised platforms to the platforms detected using the TIP method (Fig. 9). Figures 9a to f show that a left-hand tail is present for the digitised platforms, whereas platforms detected by TIP show a sharp decrease in the pdf at these elevations: this indicates the presence of more false negatives than false positives at the lowest elevations of the marsh platform. This suggests that the TIP method excludes more features with a low elevation than manual digitisation, which correspond to tidal creeks and sunken terraces at the edge of the platform. However, this does not imply that the TIP method cannot identify multiple terraces within a platform, as shown by the multiple local maxima in the detected pdf in Fig. 9d and f.

We also show maps of the TIP method's performance for each test site in order to explore this spatial variability in feature detection (Fig. 10). For instance, the dominance of false positives over false negatives in Fig. 10a (site S1) suggests that the method tends to overestimate the extent of jagged, low-relief marsh platforms, which are common in the sheltered microtidal bays characterising this site. This is the product of two factors: (i) identified scarps are not always complete in micro-tidal environments, as scarps tend to be small and therefore liable to elimination by our elevation threshold (see Fig. 4, step 4.5); and (ii) the reverse dispersion process (see Fig. 6) is then likely to encroach on the tidal flat. This phenomenon is exacerbated by coarse grids or de-noised datasets (e.g. Fig. 8a1 and a2) where high slope values are smoothed and filtered out in the scarp detection process. In our meso- to macro-tidal sites S2 to S4 (Fig. 10b-d), the method results in false negatives corresponding to the location of tidal creeks. These creeks were purposefully included in the marsh platform during the digitisation process, but were identified as part of the tidal flat by the TIP method. This result indicates that our method often characterises creek banks as platform scarps due to their morphological similarity.

Other coastal landforms may generate false positives, as seen in Fig. 10 c-f. In these cases, the position of the scarp line differs between the digitised and the TIP-detected platforms due to elevated portions of the tidal flat being adjacent to the marsh platform. This suggests that some areas of the tidal flat are topographically closer to the platform than to the rest of the tidal flat and may represent areas likely to be colonised by pioneer vegetation, even though they might not be vegetated at the time of data acquisition. Conversely, sunken platforms or fallen blocks that are not delineated by scarps may generate false negatives, as seen in the central area of Fig. 10e.

Although the TIP method was tested using salt marshes located in England, the scarp and platform association is a common feature to many salt marshes around the world, making the TIP method applicable over a wide range of geographic areas. Furthermore, the TIP method does not require the precise topography of the platform to function, making it relatively insensitive to unequal removal of vegetation between different DEM sources. The presence of vegetation induces positive errors in the DEM, which counter-intuitively may be useful when applying the TIP method, as this artificially increases the platform height and therefore the scarp slope. Examples of sites outside the United Kingdom are included in Fig. B2, and were selected to demonstrate the versatility but also the limits of the TIP method.

## 4.2 Future developments

As discussed in Section 4.1, the TIP method currently excludes tidal creeks from the marsh platform, leading to discrepancies when compared to manual digitisation. Therefore, we would expect the TIP method to underperform on highly dissected marsh platforms. As a proxy for the dissection of the platform by tidal creeks, we digitise tidal creek centrelines from the DEM. We then calculate the total length of tidal creeks included in the digitised platform divided by the platform surface area. We refer to this quantity as the Dissection Index (DI). In Fig.11, we examine the capacity of the TIP-method to determine the area and perimeter of marsh platforms according to their dissection index. We find that for all test sites, TIP-detected area remains within 10% of the digitised area, whereas TIP-detected perimeter increases steadily with Dissection Index, confirming that the exclusion of tidal creeks by the TIP method is consistently stricter than by digitisation. However, neither the TIP method nor manual digitisation offer an objective solution to detect tidal creeks. For a comprehensive analysis of marsh platforms, we recommend that objective platform detection be used in conjunction with objective creek detection methods such as those developed by Fagherazzi et al. (1999) and Liu et al. (2015). Furthermore, future developments of the TIP method will include an objective creek detection method adapted from these publications, as well as channel network extraction methods developed for fluvial channels by Clubb et al. (2014), to ensure that tidal creeks are detected as separate objects.

The morphological characteristics of prograding marshes are different from those of established platforms: consequently, vegetation patches and pioneer zones are not the object of the TIP method. Specifically, prograding margins and vegetation patches tend to have a relief and slope that are close to those of the tidal flat, making their outlines invisible to the scarp routing process. The combined absence of scarps and low relief of prograding marshes then interfere with the 20 cm leeway included in the platform filling process and cause an excess of false positives. Users may reduce this leeway to improve accuracy (see Fig. B2b1), but we discourage the use of the TIP method to identify vegetation patches and prograding margins. However, these dynamic features are the centrepiece of salt marsh development and would benefit from reproducible monitoring methods. Future research may build on the works of Balke et al. (2012) to determine characteristic morphologies of prograding marshes, thus providing the necessary groundwork to enable reproducible monitoring.

## 4.3 Potential for monitoring

As well as providing us with the ability to automate the delineation and analysis of marsh platforms across multiple sites, our method also allows the objective detection of change in marsh extent through time, with important implications for habitat monitoring or carbon storage evaluation. We test the capacity of the TIP method to monitor temporal change through the example of site S6, which was affected by heavy rainfall in the summer of 2007, resulting in high discharge in rivers such as the Parrett. 1 m lidar data distributed by the Environment Agency shows that between March and October 2007 the North-Eastern corner of site S6 underwent significant erosion. Blue pixels indicating loss of elevation (between March and October) in Fig. 12a bear the characteristic shape of slope failures and intersect the both the automatically- and manually-detected platform outline of March 2007, showing that the October platform outline is further inland.

This retreat of the marsh platform is observed both by the objectively classified (Fig. 12b) and the manually digitised platforms (Fig. 12c). However, whereas the digitisation effort focuses on the large bank failures, the TIP method also detects small changes in the DEM at the platform margin (visible in Fig. 12a and b), and may detect them as changes in marsh platform extent. Consequently, despite a close correspondence between TIP-determined marsh outlines and digitised outlines (Fig. 12a) near the bank failures, the digitised volume loss is only 81% of the objectively detected volume loss. Pioneer zones, characterized by shallow slopes and rapid, uneven elevation changes, are also likely to generate small topographic differences between the DEMs.

## 5  Conclusions

In this study we have presented a novel method which uses the topographic signature of salt marsh platforms to determine their seaward extent on high resolution DEMs. By combining non-dimensional search parameters and empirical calibration, it separates marsh platforms from tidal flats with over 90% accuracy for source data of up to 3 m in grid resolution, a result sufficient to allow quantitative morphology analyses and monitoring, particularly for eroding marshes where scarps are clearly defined. Independence from environmental variables means that our method can be used to complement spectral data for identifying plant types, to better understand feedbacks between sedimentation, deposition and biomass. We tested our method on six sites with a wide range of spring tidal ranges and found that tidal range has no significant impact on the detection accuracy. Furthermore, the presence of algae, kelp or duckweed as well as varying vegetation reflectance properties, which may induce specific calibrations with spectral methods (Morris et al., 2005), do not affect our results (barring mounds of stranded algae large enough to affect topography). Although we did not test the performance of the TIP method on DEM resolutions finer than 1 m, the option of applying a Wiener filter to reduce DEM noise is available to accommodate DEMs generated from unclassified point clouds, which have higher surface roughness. When combined with creek detection methods, we expect the performance of the TIP method to improve with fewer false negatives. This would also allow the discrimination of channel evolution within the marsh platform and on the tidal flat, allowing us to simultaneously explore the development of marsh platforms and tidal creeks (D'Alpaos et al., 2007, 2010) in sites with strong tidal forcing.

Furthermore, the unsupervised detection of marsh platforms from their topography alone reduces the computational cost of topographic analysis compared to spectral studies. This promotes the consideration of salt marshes as topographic objects as well as ecological systems, facilitating holistic, data-driven studies on salt marsh eco-geomorphic responses, and testing existing models of eco-geomorphic feedback (e.g. Fagherazzi et al., 2012). It also encourages us to think of the topographic object separately from the ecological system: mismatches in their respective boundaries may therefore be used to investigate accretion processes and pioneer zone growth in continuation with the works of Balke et al. (2014) and Hu et al. (2015). The examination of such processes at smaller scales, such as those obtained with terrestrial lidar stations, may also reveal characteristic accretion patterns (Balke et al., 2012) which topographic methods may objectively detect. Other developments of this method may, in time, enable the detection of the spatial extent of other ecosystems, such as riparian wetlands and mangrove limits.

*Code and data availability.* Our software is freely available for download on GitHub as part of the Edinburgh Land Surface Dynamics Topographic Tools package at https://github.com/LSDtopotools. The software used in this study is available in this release: https://github.com/LSDtopotools/LSDTopoTools_MarshPlatform/releases/tag/v0.2 (Goodwin et al., 2017).

*Author contributions.* GCHG designed the method with contributions from other authors. GCHG wrote the code and produced the figures, with support from SMM and FJC in integrating methods with existing channel extraction and topographic processing algorithms. GCHG wrote the paper with contributions from other authors.

*Competing interests.* The authors declare no competing interests.

*Acknowledgements.* GCHG was supported by a NERC doctoral training partnership grant (NE/L002558/1). SMM was supported by the Leverhulme Foundation (IAF-2014-009). FJC was supported by NERC grant NE/P012922/1. The authors acknowledge the United Kingdom Environment Agency for the consequent amount of lidar data (point cloud and gridded) made freely available through their website. The authors thank Dr. Dimitri Lague for his insightful comments.

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

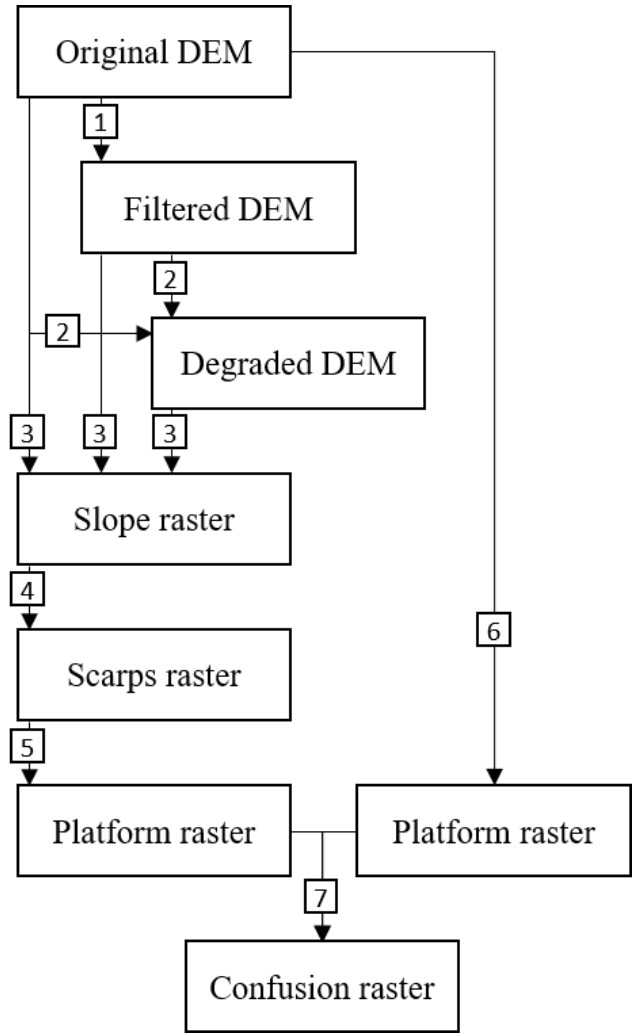

**Figure 1.** Flow chart showing the overall structure of the TIP method and its validation. Each object (rectangle) is obtained by implementing a routine (square), numbered as follows: 1. Implementation of a Wiener filter (optional); 2. Subsampling by average value (optional); 3. Calculation of slope by fitting a second order polynomial surface; 4. Scarp identification by routing; 5. Platform identification by dispersion; 6. Manual digitisation of a marsh platform; 7. Comparison of the objectively detected platform to the manually digitised platform.

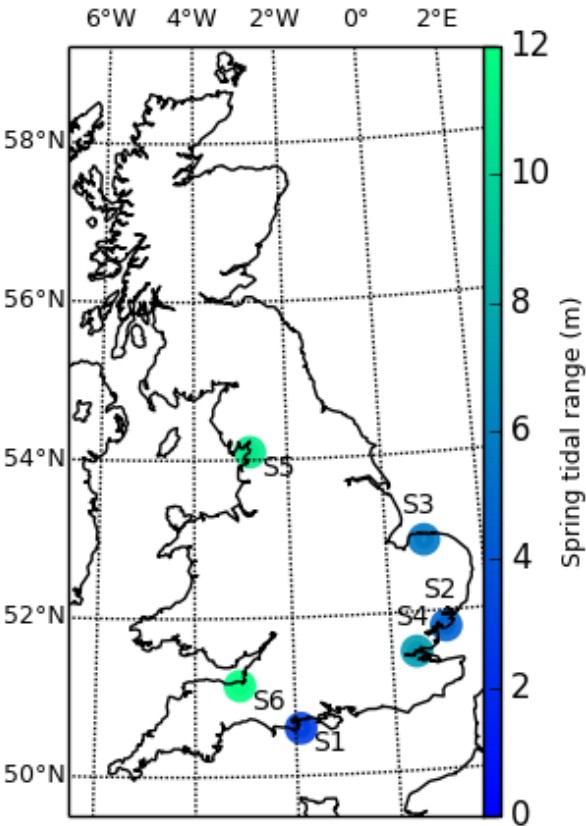

**Figure 2.** This map shows the six sites selected from the lidar collection of the UK environment agency, coloured by spring tidal range. The sites are numbered as follows: S1: Shell Bay, Dorset; S2: Stour Estuary, Suffolk; S3: Stiffkey, Norfolk; S4: Medway Estuary,Kent; S5: Jenny Brown's Point, Lancashire; S6: Parrett Estuary, Somerset.

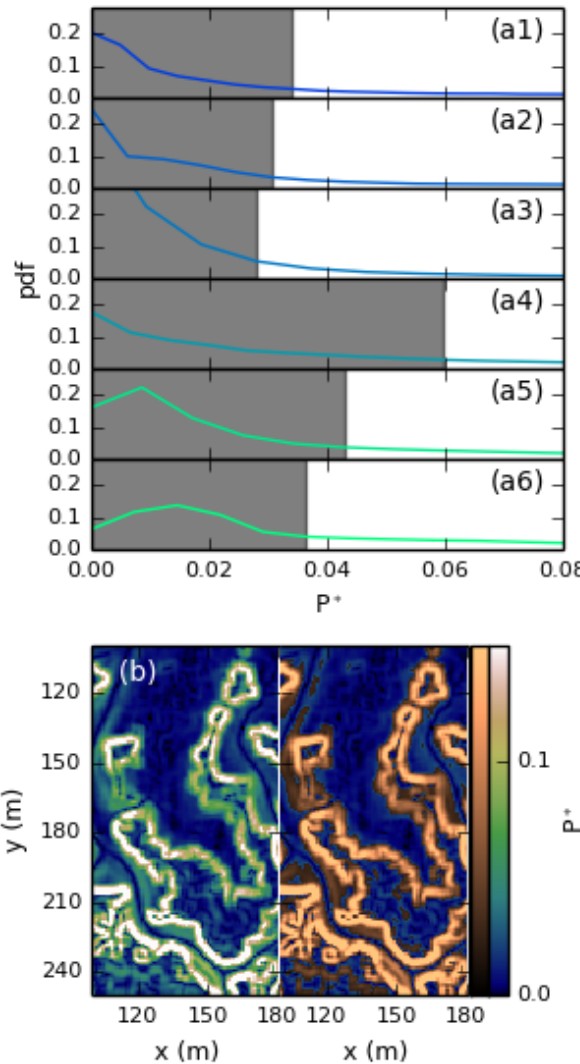

**Figure 3.** a1-6. Frequency distribution of $P^*$ for sites S1-6. The greyed portion of the plot represents pixels that are not included in the initial search space $Ss_1$; b. raster representation of $P^*$ for site S1: Shell Bay. Values of $P^*$ under $P^*_{th}$ use the topographic colour scheme, while values above $P^*_{th}$ use the copper colour scheme and are included in $Ss_1$.

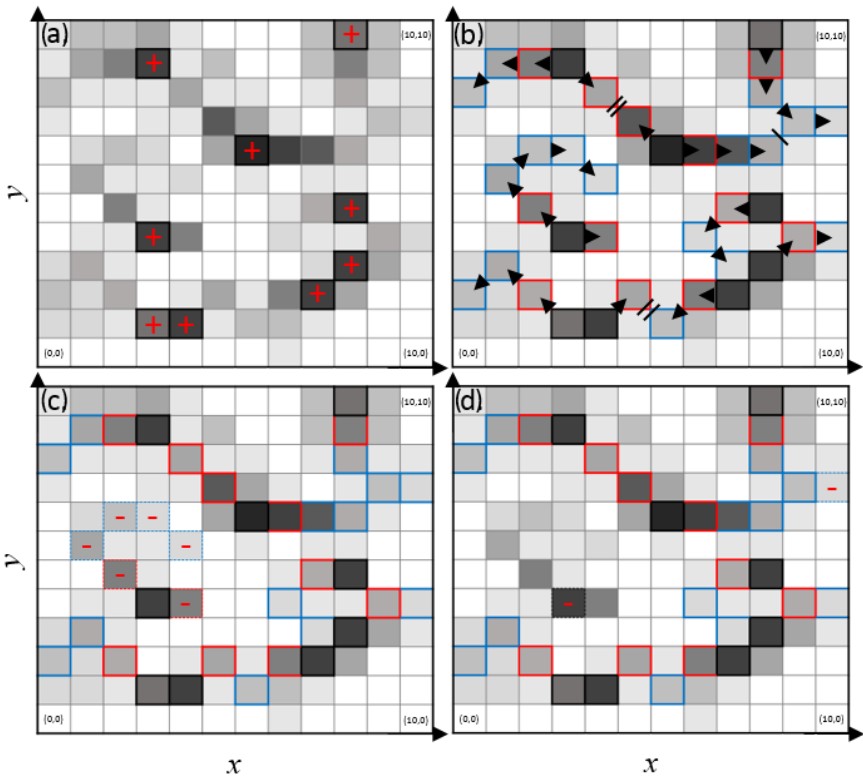

**Figure 4.** Schematic example of the scarp detection process through maximum slope routing. Panel a. shows two steps. Step 4.1: determination of the search space $Ss_1$ (greyed cells, darker with arbitrary slope). Step 4.2: Determination of local maxima $Sc_1$ (black outlines with a plus sign); b. Step 4.3: Determination of $Sc_2$ cells (red outlines). Step 4.4: Determination of $Sc_n$ cells, $n>2$ (blue outlines); c. Step 4.5: Elimination of cells where $max(Zk_9) < 0.85 \times q_{75}$ (dashed outlines with a minus sign); d. Step 4.6: Elimination of isolated cells (dashed outlines with a minus sign). The arrows represent the progressive selection of scarp cells.

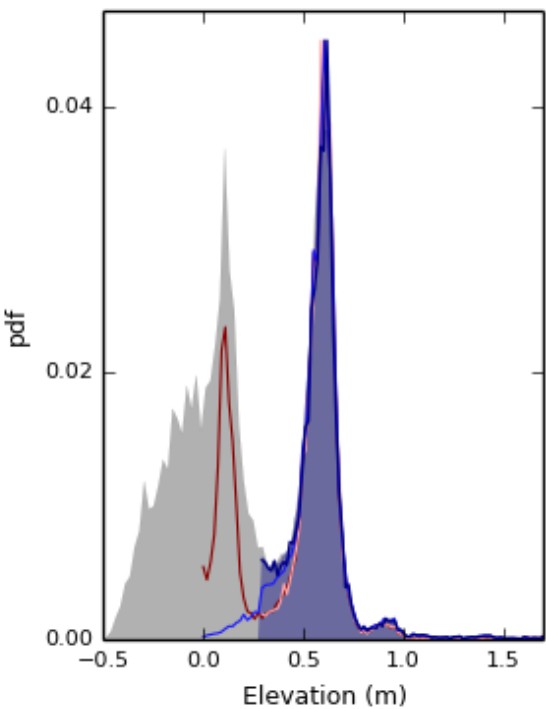

**Figure 5.** Diagram describing the elimination of the tail of the elevation probability distribution function for site S1. The grey filled surface is the pdf of elevation for the original DEM. The dark red line is the pdf of elevation of the platform after the dispersion process. The orange line is the pdf of elevation of the platform after truncation of the tail of the distribution. The blue line is the pdf of elevation of the platform after filling pools and jagged outlines and after the addition of scarps in the platform ensemble. The dark blue line, associated to the blue filled surface, is the pdf of elevation for the final platform, after the tail of its distribution is truncated a second time. All distributions in this plot are forced to display the same maximum for clarity.

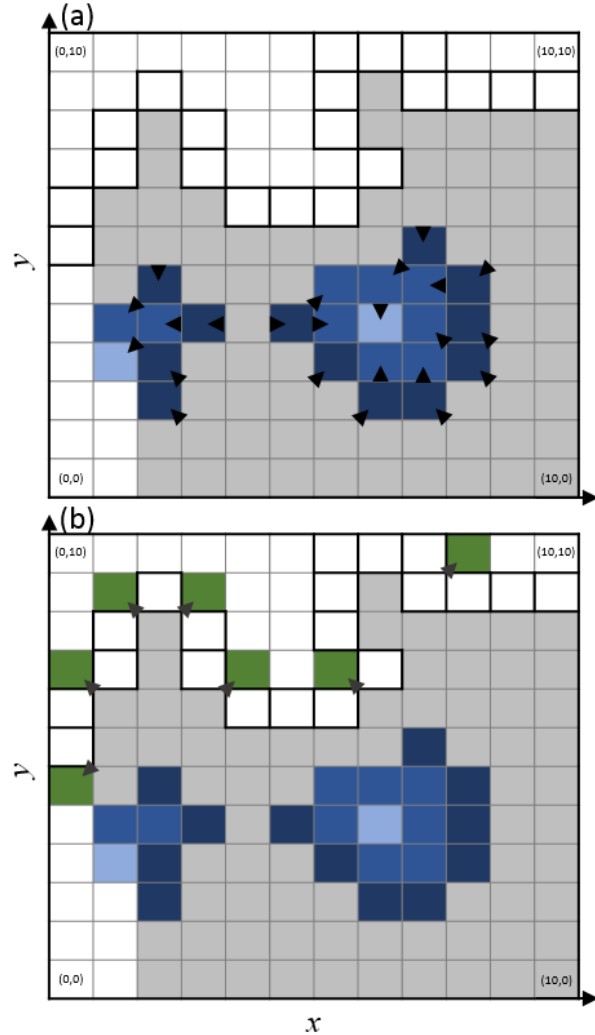

**Figure 6.** Schematic example of the reverse platform filling process. a. Step 5.1: Filling of empty cells adjacent to $Pc_n$ cells (grey, dark blue and blue cells) with and order $n-1$ (dark blue, blue and light blue cells); b. Step 5.2: Filling of empty cells adjacent to $Pc_n$ cells (grey cells) with and order $n-1$ (green cells) when scarp cells (black outlines) are included in the platform ensemble. The arrows indicate the dispersion pattern.

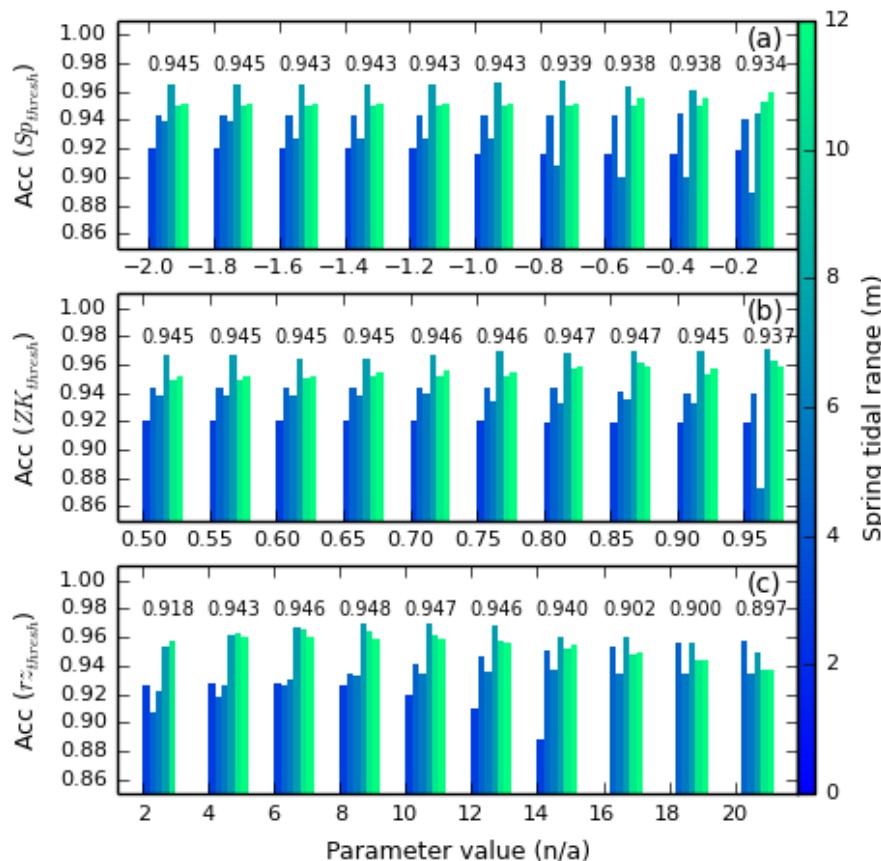

**Figure 7.** Accuracy charts used to optimize the three user-defined parameters for the six test sites, each site being coloured by spring tidal range, with no filter. Each group of bars represents the accuracy for one parameter value when applied to all the test sites. The mean accuracy appears above each group; a. Accuracy for the parameter Opt1. The retained value for Opt1 is -2.0; b. Accuracy for the parameter Opt2. The retained value for Opt2 is 0.85; c. Accuracy for the parameter Opt3. The retained value for Opt3 is 8.

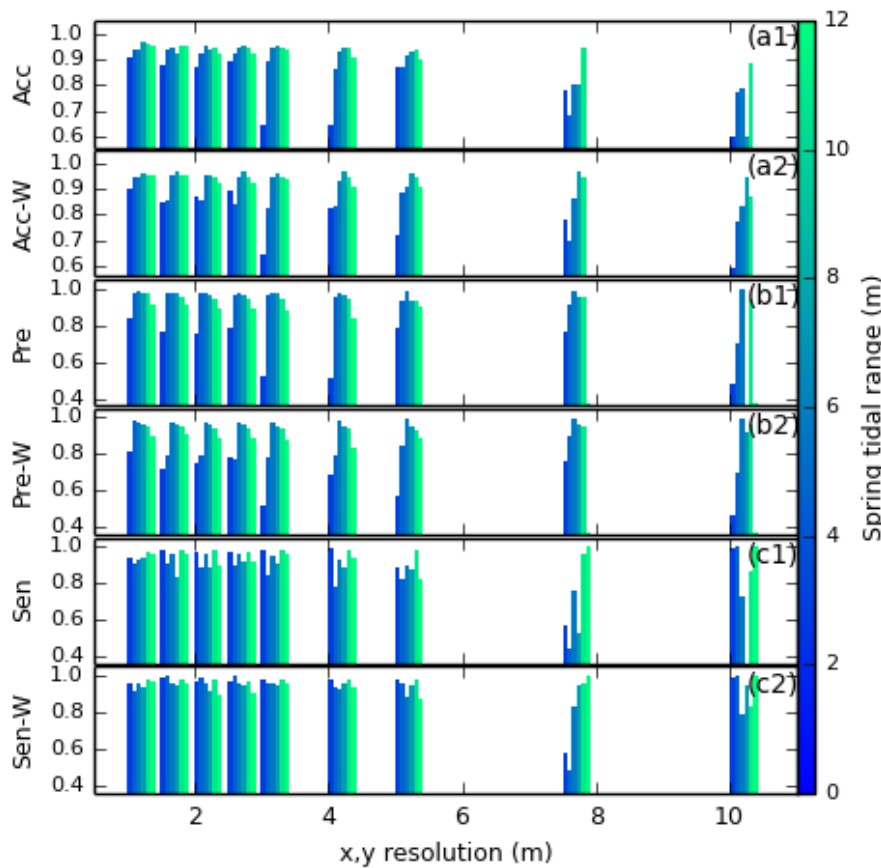

**Figure 8.** Performance of the platform detection method for all sites, coloured according to their spring tidal range; a1. Accuracy of the method when no filter is used; a2. Accuracy of the method when using a Wiener filter; b1. Precision of the method when no filter is used; b2. Precision of the method when using a Wiener filter; c1. Sensitivity of the method when no filter is used; c2. Sensitivity of the method when using a Wiener filter.

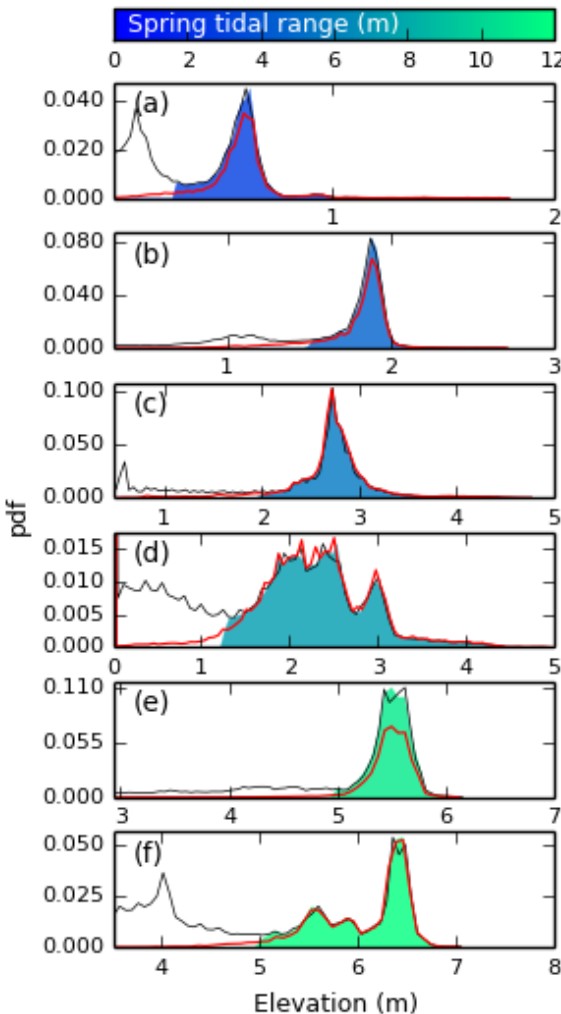

**Figure 9.** Elevation distribution functions for sites S1 to S6 (plots a. to f. respectively). The red line corresponds to the elevation distribution for the reference rasters. The filled area corresponds to the elevation distribution of the automatically processed rasters, coloured according to their spring tidal range. The grey line represents the elevation distribution of the original DEM, with frequency maxima set to match those of the automatically processed rasters so as to nullify the effect of empty cells.

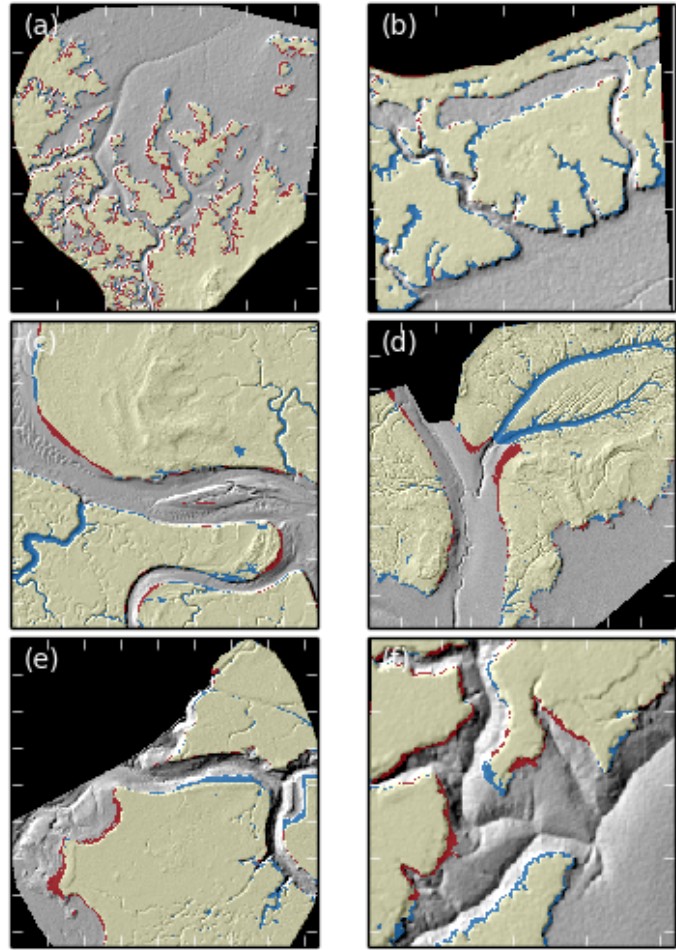

**Figure 10.** Rasters comparing digitised versus extracted marsh platforms superimposed on hillshade data for all six sites after detection with no Wiener filtering. Black areas are outside of the detection domain and contain no data. Yellow areas correspond to True Positives (TP) and transparent areas to True Negatives (TN). Red areas correspond to False Positives (FP) and blue areas to False Negatives (FN). Ticks are placed 50m apart. The sites are numbered as follows: a: Shell Bay, Dorset; b: Stour Estuary, Suffolk; c: Stiffkey, Norfolk; d: Medway Estuary,Kent; e: Jenny Brown's Point, Lancashire; f: Parrett Estuary, Somerset.

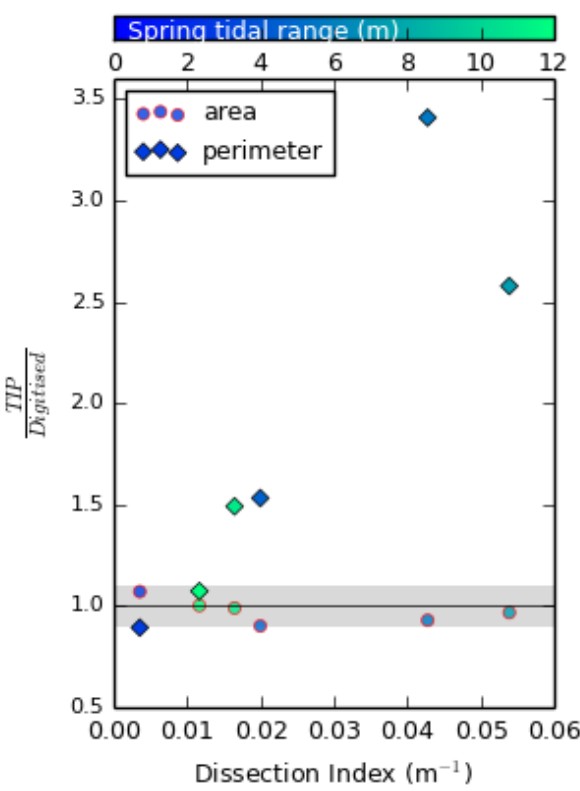

**Figure 11.** Ratio of TIP over digitised area (circles, red outlines) and perimeter (diamonds, black outlines) for sites S1 to S6 at the native resolution of 1 m, with no Wiener filtering, as a function of dissection index. Here, dissection index is defined as the ratio of the total length of tidal channels within the digitised marsh platform over the area of the digitised marsh platform, and is not bounded by drainage basins. The greyed area corresponds to a 10% buffer around the line of equation *y=1*.

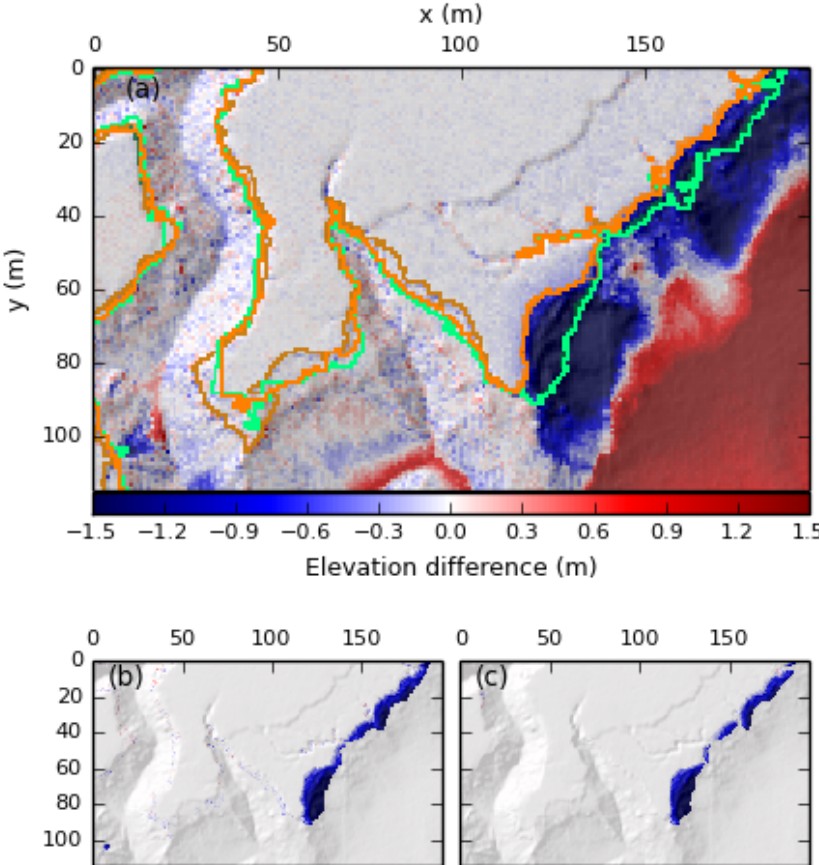

**Figure 12.** a. Comparison of marsh areas for a portion of S6 between March (green lines) and October (orange lines) 2007, surperimposed on hillshade data of October 2007. Bright lines correspond to the automatically detected marsh boundary, whereas faded lines correspond to digitised marsh boundaries. Green faded lines are mostly covered by bright green lines. Coloured surfaces indicate elevation gain or loss between March and October 2007; b. Map of elevation loss and gain associated to marsh platform evolution, according to the TIP method. Total volume loss is 1188 $m^3$; c. Map of elevation loss and gain associated to marsh platform evolution, according to manual digitisation. Total volume loss is 966 $m^3$.

**Appendix A:  TIP performance tables**

|      | S1    | S2    | S3    | S4    | S5    | S6    |
|------|-------|-------|-------|-------|-------|-------|
| 1.0  | 0.907 | 0.94  | 0.936 | 0.967 | 0.963 | 0.952 |
| 1.5  | 0.876 | 0.934 | 0.948 | 0.926 | 0.953 | 0.95  |
| 2.0  | 0.868 | 0.921 | 0.95  | 0.942 | 0.945 | 0.919 |
| 2.5  | 0.891 | 0.926 | 0.948 | 0.955 | 0.942 | 0.926 |
| 3.0  | 0.646 | 0.897 | 0.944 | 0.954 | 0.946 | 0.935 |
| 4.0  | 0.643 | 0.861 | 0.932 | 0.942 | 0.945 | 0.909 |
| 5.0  | 0.869 | 0.872 | 0.915 | 0.927 | 0.941 | 0.897 |
| 7.5  | 0.778 | 0.682 | 0.804 | 0.806 | 0.942 | 0.376 |
| 10.0 | 0.599 | 0.771 | 0.786 | 0.603 | 0.882 | 0.376 |

**Table A1.** Table of Accuracy for sites S1 to S6 (columns) with no Wiener filter, for resolutions varying between 1 and 10 m (rows).

|      | S1    | S2    | S3    | S4    | S5    | S6    |
|------|-------|-------|-------|-------|-------|-------|
| 1.0  | 0.837 | 0.979 | 0.985 | 0.972 | 0.973 | 0.916 |
| 1.5  | 0.763 | 0.97  | 0.977 | 0.974 | 0.952 | 0.91  |
| 2.0  | 0.753 | 0.971 | 0.976 | 0.967 | 0.941 | 0.89  |
| 2.5  | 0.789 | 0.961 | 0.976 | 0.969 | 0.942 | 0.889 |
| 3.0  | 0.518 | 0.959 | 0.975 | 0.974 | 0.943 | 0.88  |
| 4.0  | 0.513 | 0.951 | 0.977 | 0.968 | 0.942 | 0.835 |
| 5.0  | 0.787 | 0.936 | 0.989 | 0.932 | 0.932 | 0.896 |
| 7.5  | 0.765 | 0.908 | 0.988 | 0.956 | 0.949 | 0.376 |
| 10.0 | 0.475 | 0.699 | 0.992 | 0.0   | 0.947 | 0.376 |

**Table A2.** Table of Precision for sites S1 to S6 (columns) with no Wiener filter, for resolutions varying between 1 and 10 m (rows).

|      | S1    | S2    | S3    | S4    | S5    | S6    |
|------|-------|-------|-------|-------|-------|-------|
| 1.0  | 0.94  | 0.913 | 0.931 | 0.943 | 0.973 | 0.962 |
| 1.5  | 0.981 | 0.91  | 0.956 | 0.834 | 0.981 | 0.963 |
| 2.0  | 0.974 | 0.883 | 0.959 | 0.882 | 0.981 | 0.895 |
| 2.5  | 0.972 | 0.902 | 0.956 | 0.916 | 0.975 | 0.915 |
| 3.0  | 0.985 | 0.849 | 0.953 | 0.906 | 0.98  | 0.956 |
| 4.0  | 0.992 | 0.786 | 0.934 | 0.882 | 0.979 | 0.945 |
| 5.0  | 0.892 | 0.821 | 0.901 | 0.88  | 0.984 | 0.823 |
| 7.5  | 0.571 | 0.448 | 0.757 | 0.533 | 0.965 | 1.0   |
| 10.0 | 0.996 | 1.0   | 0.731 | nan   | 0.87  | 1.0   |

**Table A3.** Table of Sensitivity for sites S1 to S6 (columns) with no Wiener filter, for resolutions varying between 1 and 10 m (rows).

|      | S1    | S2    | S3    | S4    | S5    | S6    |
|------|-------|-------|-------|-------|-------|-------|
| 1.0  | 0.9   | 0.943 | 0.948 | 0.961 | 0.95  | 0.948 |
| 1.5  | 0.847 | 0.857 | 0.948 | 0.963 | 0.953 | 0.95  |
| 2.0  | 0.868 | 0.854 | 0.95  | 0.956 | 0.945 | 0.919 |
| 2.5  | 0.89  | 0.838 | 0.948 | 0.964 | 0.942 | 0.923 |
| 3.0  | 0.646 | 0.828 | 0.947 | 0.962 | 0.945 | 0.935 |
| 4.0  | 0.824 | 0.832 | 0.931 | 0.964 | 0.945 | 0.91  |
| 5.0  | 0.717 | 0.882 | 0.904 | 0.961 | 0.941 | 0.91  |
| 7.5  | 0.777 | 0.698 | 0.864 | 0.965 | 0.942 | 0.376 |
| 10.0 | 0.593 | 0.771 | 0.833 | 0.945 | 0.87  | 0.376 |

**Table A4.** Table of Accuracy for sites S1 to S6 (columns) with a Wiener filter, for resolutions varying between 1 and 10 m (rows).

|       | S1    | S2    | S3    | S4    | S5    | S6    |
|-------|-------|-------|-------|-------|-------|-------|
| 1.0   | 0.816 | 0.978 | 0.976 | 0.963 | 0.948 | 0.9   |
| 1.5   | 0.716 | 0.798 | 0.977 | 0.961 | 0.952 | 0.91  |
| 2.0   | 0.753 | 0.795 | 0.976 | 0.966 | 0.941 | 0.89  |
| 2.5   | 0.787 | 0.774 | 0.976 | 0.962 | 0.942 | 0.889 |
| 3.0   | 0.518 | 0.778 | 0.976 | 0.951 | 0.944 | 0.88  |
| 4.0   | 0.687 | 0.794 | 0.979 | 0.948 | 0.943 | 0.841 |
| 5.0   | 0.571 | 0.846 | 0.993 | 0.953 | 0.932 | 0.887 |
| 7.5   | 0.757 | 0.897 | 0.99  | 0.962 | 0.951 | 0.376 |
| 10.0  | 0.471 | 0.699 | 0.995 | 0.919 | 0.96  | 0.376 |

**Table A5.** Table of Precision for sites S1 to S6 (columns) with a Wiener filter, for resolutions varying between 1 and 10 m (rows).

|      | S1    | S2    | S3    | S4    | S5    | S6    |
|------|-------|-------|-------|-------|-------|-------|
| 1.0  | 0.955 | 0.92  | 0.957 | 0.938 | 0.982 | 0.971 |
| 1.5  | 0.993 | 0.997 | 0.956 | 0.945 | 0.981 | 0.963 |
| 2.0  | 0.974 | 0.993 | 0.959 | 0.92  | 0.982 | 0.895 |
| 2.5  | 0.973 | 0.999 | 0.956 | 0.946 | 0.975 | 0.909 |
| 3.0  | 0.985 | 0.961 | 0.955 | 0.953 | 0.977 | 0.956 |
| 4.0  | 0.976 | 0.936 | 0.931 | 0.961 | 0.979 | 0.938 |
| 5.0  | 0.978 | 0.958 | 0.883 | 0.948 | 0.985 | 0.873 |
| 7.5  | 0.581 | 0.489 | 0.834 | 0.95  | 0.964 | 1.0   |
| 10.0 | 0.996 | 1.0   | 0.79  | 0.946 | 0.838 | 1.0   |

**Table A6.** Table of Sensitivity for sites S1 to S6 (columns) with a Wiener filter, for resolutions varying between 1 and 10 m (rows).

## Appendix B: Additional test sites and limitations of the TIP method

Here we present three additional sites that demonstrate the capabilities and limits of the TIP method. Sites were selected based on the availability of gridded 1 m DEMs on OpenTopography (http://www.opentopography.org) and on the variety of tidal ranges and climates present: we analyse Morro Bay, CA (A1), Wax Lake Delta, LA (A2) and Plum Island, MA (A3, see Fig. B1). As is common of marshes in the United States, these additional sites have a lower relief than many European marshes, with site A2 displaying a relief of 0.8 m. The performances of the TIP method are recorded in Fig. B2. Optimisation parameters were maintained within the ranges described in Fig. 7.

Site A1, located in the North-East of Morro Bay, shows an extremely close correspondence between the digitised and TIP-detected platforms, with an accuracy of 97%. It also demonstrates the ability of the TIP method to detect marsh platforms in DEMs where tidal flats exist at higher elevations, as shown by the similar and non-null probability of the TIP-detected and digitised platforms at elevations between 0.3 and 0.9 m (Fig. B2b1). To confirm the observations drawn in the body of the article, site A1 displays an abundance of false negatives within tidal creeks (Fig. B2a1), adding weight to the argument that these features require independent treatment.

Site A2 is located on the inside of a marsh island in the rapidly growing Wax Lake Delta. In order to detect the marsh platform with the performance reported in Fig. B2b2, the minimum elevation buffer of 20 cm used in step 5 of Fig. 1 to fill marsh platforms was reduced to 5 cm. This allows the TIP method to function in a site with very low relief and poorly defined scarps. However, we note in Fig. B2b1 that the marginal patches of the marsh are not well identified by the method, as indicated by the relatively large number of false positives on the outline of the marsh. This example therefore demonstrates the difficulties experienced when attempting to detect a prograding marsh by the TIP method. We therefore recommend caution when using the TIP method to monitor prograding marshes, as additional work is needed to fully characterise the topographic signatures of fallen blocks and pioneer zones.

Site A3 is a portion of the well-studied Plum Island, MA. The TIP method yields similar results to site A1, with the notable exception of the bottom right corner of Fig. B2c1. In this area, the marsh platform is heavily dissected by wide, shallow pools and channels, which are commonly excluded from the platform ensemble by the TIP method. Furthermore, the excluded area (containing most false negatives) forms a low, shallow concave surface within the marsh, typically associated with seasonally vegetated areas. These features are morphologically similar to a high tidal flat within the platform, and are therefore difficult to identify using the TIP method.

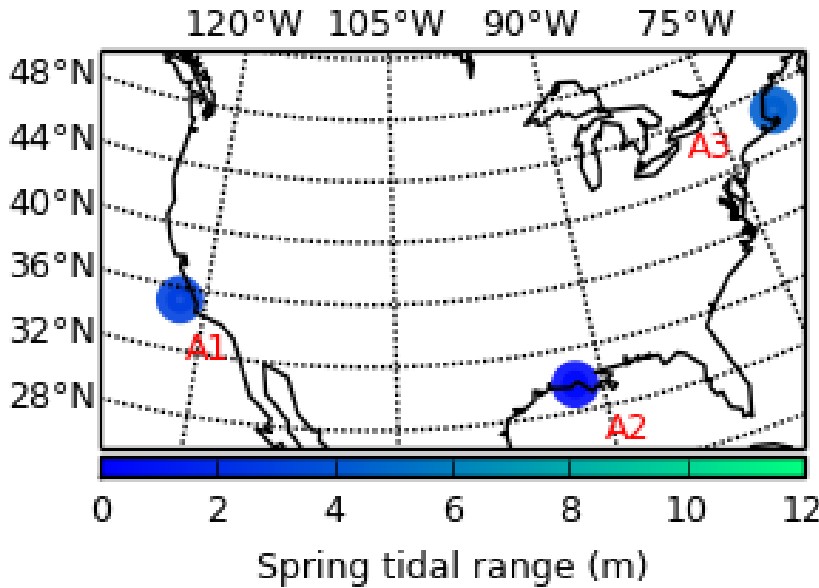

**Figure B1.** This map shows the three additional sites selected from the lidar collection of OpenTopography (http://www.opentopography.org), coloured by spring tidal range. The sites are numbered as follows: A1: Morro Bay, California; A2: Wax Lake Delta, Louisiana; A3: Plum Island, Massachusetts.

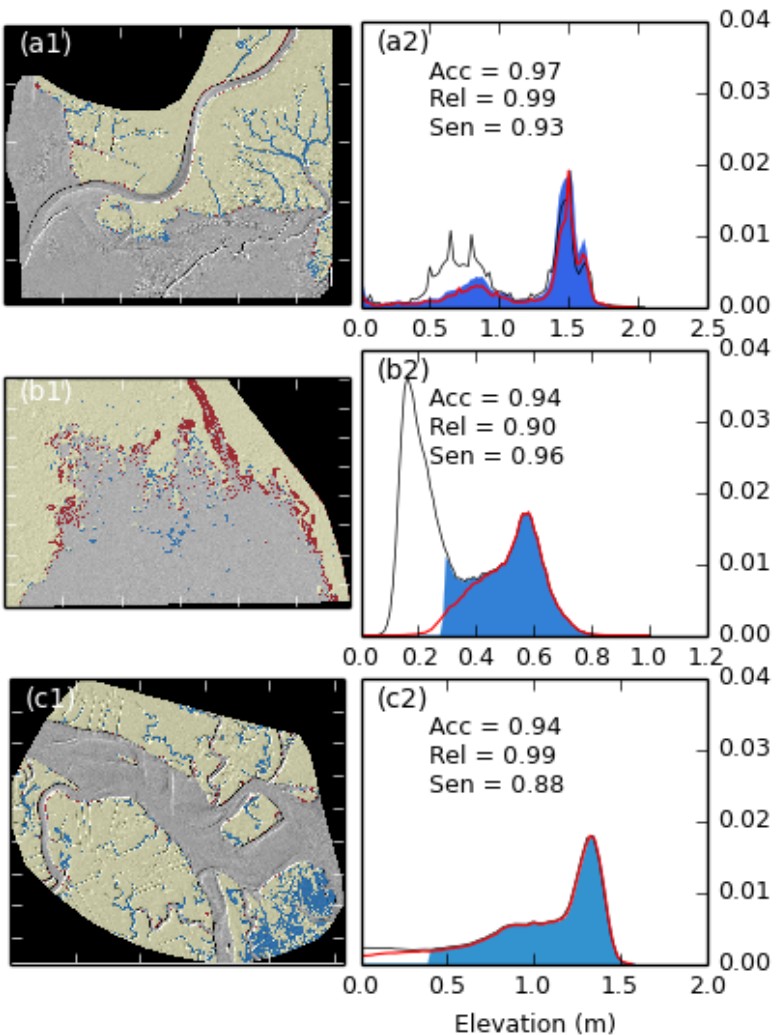

**Figure B2.** This figure combines the map found in Fig. 10 (a1, b1 and c1) and the probability distribution functions in Fig. 9 as well as the values of Accuracy, Precision and Sensitivity for sites A1 to A3 (a2, b2, c2). Each DEM was processed at its native resolution of 1 m.