# Peer review of "Unsupervised detection of salt marsh platforms: a topographic method"

_Earth Surface Dynamics, 2017_

## Referee Comment (RC1) · Anonymous Referee #1 · 21 Nov 2017

Overall Quality (General Comments):

The authors present a novel approach to autonomously (unsupervised) detect the spatial extent of marsh platform areas from a high resolution salt marsh DEM in a GIS environment. While the method seems somewhat useful, their approach is limited in that it is only applicable for marsh surfaces characteristic of steep scarps, as in erosional landscapes. Nevertheless, the unsupervised approach (TIP) presented by the authors is easy to follow and is repeatable, and can be very useful in change-detection analysis of natural and constructed salt marshes, particularly erosional landscapes, assuming that a high resolution DEM is available. Therefore, I think this paper can become a nice addition, pending the authors address the specific comments and technical suggestions outlined below.

[Figure]

Specific Comments:

1. I worry that the authors may underestimate the level of detail needed to accurately resolve the decimeter scale topography of the marsh platform in the requisite DEM. The authors rely heavily on widely available Lidar DEMS for the TIP method despite the fact that the overall relief of the marsh platform is often missed completely by Lidar sensors. Perhaps it doesn't really matter here since the authors are establishing marsh platform identification on the scarp perimeters...but, I wonder if there are any ways you might improve on your method to extend its usefulness to other marsh landscapes (those without scarps, and those characteristic of "patchy", discontinuous areas of marsh platform that might be heavily dissected by intertidal creek networks.

2. The method presented here is only useful in marsh landscapes characteristic of steep scarps (as in erosional environments). I think that the title should reflect that in some way.

3. Perhaps you could include more descriptive information on the geomorphology of your study sites. For instance, you have high resolution DEMs for all, why not calculate drainage density, or some other metric to describe how heavily dissected the marsh platform is? Then, your results could vary as a function of drainage density and tidal range? Maybe. . .

4. The paper could use some organizational finesse to improve the flow of the narrative. There are many instances where results are stated in the Methods section, and there is no Discussion section, but discussion elements are mixed in with Results. I would also consider adding a separate section for "Validation" following or within the Methods section to describe how you evaluated the performance of the TIP method. It seems very out of place in its current position (Results and Discussion). See specific comments below.

5. The Results section is a bit messy. Perhaps consider organizing into a more logical manner. For instance, I like the idea of presenting results as a function of tidal range (or

[Figure]

drainage density – see comment #3). . .start with S1, describe, then go on to S2. . .and so on. Then, in a separate section (see comment #4) you could demonstrate the effects of using the filter on TIP results. I think this approach would be fine, because you already told us that you don't want to use the filter. . .and that's OK.

Technical Comments:

ABSTRACT:

P1L3: The productivity and even survival of salt marsh. . .(remove "even")

P1L3: . . .of salt marsh vegetation. . .(why vegetation here? Why not landscape? Seems out of place.)

P1L5: Determining platform boundaries. . .(Determination of platform boundaries. . .)

P1L7: saltmarsh (salt marsh. . .make this change throughout. There are several instances of saltmarsh vs salt marsh.)

P1L15: . . .allows the accurate. . . (allows for the accurate)

P1L20: . . ., it also suggests. . . (what is 'it'?)

INTRODUCTION:

P2L9: awkward, consider revising. . .perhaps something like, ". . .makes monitoring the evolution of salt marshes imperative for management strategies and scientific endeavors. . .".

P2L34: Right. . .but marsh platform slopes are on the order of 30cm total relief. . .and the overall structure is often misrepresented by lidar sensors with a nominal accuracy of +/- 15cm.

P3L17: . . .horizontal resolutions. . .(do you mean horizontal extents? These are two very different things.)

P3L18: remove dash after "short".

METHODS:

P3L27: can you provide any technical specs for the lidar survey? Seasonality? Tides? Etc. . .

P4L4: stronger than what?

P4L6: . . .provides. . .(change to provide)

P4L8: what do you mean by "numerous"? How many more channels are at this site compared to the others? Consider using more physical descriptors throughout your study site description. What are the respective areas?

P4L10: What do you mean by "levels"? Elevation? Water level?

P4L19: Why three times the horizontal resolution of the DEM? Why not 5 or 6?

P4L29-32: At what scale is this problematic? 100's of kilometers? 10's of kilometers? I thought we were focused on relatively small areas of marsh landscape. . .what are the relative sizes of each study site (see also comment P4L8 above).

P5L6-20: Can you briefly describe what each of these means physically and the importance of information provided by each?

P5L23 (and throughout): be careful to avoid stating results in Methods section.

P5L24: what is "pdf"? define.

P6L20: large number of true scarps? Or do you mean large number of misidentified scarps that are actually creek banks?

P7L7: why 11?

P7L16 (and throughout): results presented in Methods section

P7L27 (and throughout): revise without the use of "we"

P8L2: . . .pansm. . .(typo? Do you mean "pans")

RESULTS:

P8L11 (throughout): Methods presented in results section. Consider providing a separate subsection in Methods for "validation" and then share results in the proper Results section.

P8L12-15: I'm guessing TP, FN, etc... are obtained from subtracting? Maybe show that in Methods.

P8L26: ...the manual digitization...did you even discuss that in your Methods section? What software was used? Scale?

P9L14: describe one figure at a time, and in chronological order.

P9L19-29: discussion in Results section. Consider revising.

P10L6-7: Isn't this simply a transition zone between marsh platform and tidal flat?

P10L8: ...saltings...why are you defining this here? You referred to "salting" earlier with no definition. Define earlier.

P10L17: ...yes, but it's limited to erosional landscapes with obvious scarps.

CONCLUSIONS:

P11L9: ...algae...why is this here?? Did you test for this or are you speculating? Maybe you could instead say that your method works independent of such environmental factors...it's implied, but not exactly tested for in this paper.

FIGURES:

Figure 3: remove duplicate axis labels. For instance, a single "pdf" on the y-axis would suffice. Same for "x (m)" on the x-axis. Why is there a negative value on right-hand y-axis? P* values range from 0-1, yes?

Figure 5: be mindful of keeping tenses consistent in your caption.

Figure 7: duplicate axis labels. . .remove.

Figure 8: maybe a table would be a nice complement to this figure?

Figure 9: why do you say "determined area"? Why not "unsupervised" or "TPI" area? Similarly, why do you say "reference" perimeter? Why not "Digitized"?

Figure 10: scale?

Figure 11: duplicate axis labels. . ..remove.

---

## Referee Comment (RC2) · Anonymous Referee #2 · 18 Dec 2017

The manuscript describes a new method for the automatic detection of salt marsh platforms and tidal flats making use of Lidar data. I believe that this topic is of general interest in the field. The manuscript is written very well, the objectives are clear, the methodology is described in details, the results are clear and easy to interpret, and the conclusions are presented very well.

My only concern is that the methodology is presented as a general tool for salt marsh and tidal flat identification, while I believe that its application is limited to the specific type of marshes presented in this study. I suggest the authors to 1) better clarify the specs of the methodology that are tightly linked to the morphological characteristics of the specific study sites in order to make aware the user of the limits in applying the methodology; 2) describe in more details the 6 study sites considered in this research underlying the specific peculiar morphological characteristics. This will allow the user/reader to decide if the methodology may be applied to a different study site. Moreover, the authors refer to a 20 cm value to be subtracted "to define the minimum local elevation for a platform pixel" (pag. 7 lines 7-8). Also in this case a more precise explanation should be included so that the reader can judge if this is a value typical of the considered study sites or can be generalized.

In summary, I suggest the publication of the manuscript with minor revisions.

Some specific suggestions for the authors are the following:

Pag. 3 lines 20-25: in the text I do not see a description of the gray area in Fig. 3a.

Pag 7 lines 7-10: Is the value 20cm applied to all the study sites? Could you please better explain how this specific value has been selected? is there a relation with the tidal excursion for example? Is this value specific for the English study sites?

Figure 12: the faded lines are difficult to see

---

## Author Comment (AC1) · 21 Dec 2017

This response was uploaded in the form of a supplement.

Please also note the supplement to this comment:
https://www.earth-surf-dynam-discuss.net/esurf-2017-60/esurf-2017-60-AC1-supplement.zip

[Figure]

NEW-ORLEANS

**TEST OF THE TIP METHOD ON PROGRADING MARSHES IN WAX LAKE DELTA**

0    100    200    300    400 m

**Fig. 1.**

BOSTON

TEST OF THE TIP METHOD ON CHANNELED
MARSHES ON PLUM ISLAND

0    100   200   300   400 m

**Fig. 2.**

---

## Author Comment (AC2) · 21 Dec 2017

This response was uploaded in the form of a supplement

Please also note the supplement to this comment:
https://www.earth-surf-dynam-discuss.net/esurf-2017-60/esurf-2017-60-AC2-supplement.zip

[Figure]

![TEST OF THE TIP METHOD ON PROGRADING MARSHES IN WAX LAKE DELTA. Left: location map near New-Orleans with aerial photo of study area outlined in red. Right: hillshade DEM with green overlay and red study area outline, scale bar 0–400 m.]

**NEW-ORLEANS**

**TEST OF THE TIP METHOD ON PROGRADING
MARSHES IN WAX LAKE DELTA**

0    100    200    300    400 m

**Fig. 1.**

[Figure]

BOSTON

TEST OF THE TIP METHOD ON CHANNELED
MARSHES ON PLUM ISLAND

0   100   200   300   400 m

**Fig. 2.**

---

## Author Response (AR1)

THE UNIVERSITY *of* EDINBURGH
**School of Geosciences**

Guillaume C.H. Goodwin
*School of Geosciences*
*University of Edinburgh*
*Drummond Street*
*Edinburgh, EH8 9XP*
*Phone: +44 (0)131 650 2537*
*Email: g.c.h.goodwin@sms.ed.ac.uk*

David Lundbek Egholm
Associate Editor, Earth Surface Dynamics

January 31, 2018

Dear Prof. Egholm,

Thank you for considering our manuscript 'Unsupervised detection of salt marsh platforms: a topographic method'. We are grateful to the reviewers for providing constructive feedback which has helped us to improve the manuscript.

The reviewers were primarily concerned with providing more evidence that the Topographic Identification of Platforms (TIP) method is applicable to a wide range of salt marsh platforms. They justly noted that the method was tested exclusively on sites within the United Kingdom which, although representative of many salt marsh environments, do not encompass (1) American salt marshes, (2) microtidal environments and (3) prograding salt marshes. We addressed these concerns by testing the TIP method on three additional sites in the United States: Morro Bay, CA, Wax Lake Delta, LA and Plum Island, MA. The performances of the TIP method on these sites are reported in Appendix B of the revised manuscript.

Another important comment made by both reviewers was that our original manuscript did not highlight the limitations of the TIP method clearly enough in relation to the design of the method. In order to respond to these comments we modified the structure of our manuscript, including an additional section dedicated to the influence of site properties on results of the TIP method. Appendix B also contains analysis of additional site morphologies that push the limits of the TIP method, as well as providing guidance on the applicability of our method when analysing salt marshes in challenging environments.

Please find below detailed responses to the individual points raised by each of the reviewers, along with a version of our manuscript highlighting the changes we have made to answer the reviewer comments. We have formatted *reviewer comments in italics*, and our responses are in normal font. Throughout our responses we refer to line numbers in our manuscript: these are the correct line numbers in the manuscript with the changes incorporated. We have endeavoured to address all concerns and return the manuscript in a publication-ready state.

Sincerely,

*Guillaume Goodwin*

Guillaume C.H. Goodwin

**Reviewer 1**

We thank the reviewer for their helpful suggestions. Below we describe how we adjusted the manuscript in the revised version in response to these comments.

*Comment 1: I worry that the authors may underestimate the level of detail needed to accurately resolve the decimeter scale topography of the marsh platform in the requisite DEM. The authors rely heavily on widely available Lidar DEMS for the TIP method despite the fact that the overall relief of the marsh platform is often missed completely by Lidar sensors. Perhaps it doesnt really matter here since the authors are establishing marsh platform identification on the scarp perimeters...but, I wonder if there are any ways you might improve on your method to extend its usefulness to other marsh landscapes (those without scarps, and those characteristic of patchy, discontinuous areas of marsh platform that might be heavily dissected by intertidal creek networks.*

The reviewer is entirely correct to point out that some scarp heights may be lower than the vertical accuracy of the lidar data. This resolution and the relief plays a role in selection of the minimum scarp height: please see our response to reviewer 2. We included more sites in the appendix that push the limits of the method (e.g. in very low relief landscapes). We have used the method on the Wax Lake Delta in Louisiana, and the method can detect the marsh where scarps are apparent despite the fact that the maximum relief of the point cloud is 80 cm (including the returns from vegetation).

We do want to point out that our test sites have used widely available lidar DEMs: in our test cases the method works well, and the scarps/platforms are correctly delineated by the algorithm. As suggested by the reviewer, the precise topography of the platform is not necessary for the TIP method to function, as our method is focused on detecting scarps and filling the platforms at areas of higher elevation, rather than relying on the elevations of the platform itself. This has the effect of making the TIP method less sensitive to unequal removal of vegetation between different DEM sources. In response to comments below we included a few more sites with smaller tidal ranges in the appendix, and added more cautionary language about the use of the method. However, we would also like to make clear that the method can work on microtidal marshes as long as there are scarps (more on that point later). Three examples of American salt marshes were added in Appendix B of the revised manuscript.

We chose not to include patch detection or tidal creek detection in the TIP method for this manuscript. This requires the implementation of different algorithms as they have distinct morphological characteristics, and we feel this is a different topic. We agree with the reviewer that such an algorithm would be very beneficial, but we feel this is beyond the scope of the current manuscript. We have, however, tested the TIP method on expanding patches (see the results for Wax Lake Delta in Appendix B of the revised manuscript).

*Comment 2: The method presented here is only useful in marsh landscapes characteristic of steep scarps (as in erosional environments). I think that the title should reflect that in some way.*

The term 'platform' in the title is meant to reflect the necessity of the presence of a scarp in our method. We added the following definition: 'We here define salt marsh platforms as sub-horizontal surfaces in the coastal landscape, separated from surrounding intertidal flats by steep scarp features.' on P3L9 to clarify this point.

We do feel this comment suggests the method is somehow a niche method only applicable to limited settings. The authors have personally conducted field campaigns across marshes in northern France (in a macrotidal environment), in northern Italy (in microtidal environments), across the UK (along a range of sea level rise rates and tidal ranges), along the Atlantic coast of the United States

(South Carolina and North Carolina) and in the Gulf Coast of Florida. We have also recreationally visited marshes in Louisiana, California and Oregon. In all cases these marshes had platforms and scarps, despite the wide variety in vegetation species, tidal range, sea level rise rates, suspended sediment concentrations, temperatures, and wave climates. We acknowledge not all marshes have scarps and therefore not all marshes are amenable to the TIP approach, but we do wish to emphasise that this method should be broadly applicable over a wide range of geographic areas. In Appendix B we added three sites outside of the UK to demonstrate the method is not limited to the 6 specific sites we chose for intensive method verification. These places were also chosen to highlight places where the algorithm is unlikely to work to ensure that readers are aware of any pitfalls.

**Comment 3**: *Perhaps you could include more descriptive information on the geomorphology of your study sites. For instance, you have high resolution DEMs for all, why not calculate drainage density, or some other metric to describe how heavily dissected the marsh platform is? Then, your results could vary as a function of drainage density and tidal range? Maybe. . .*

We agree with the reviewer that it could be useful to look at the performance of the method in platforms with different degrees of dissection. Having published previously on drainage density (Clubb et al., 2016, JGR-ES, doi:10.1002/2015JF003747), we are slightly wary of using this specific metric. Drainage density is defined as the length of the channels in a basin divided by the basin area, but basin area in a marsh context is extremely difficult to quantify. Furthermore, many tidal channels in marsh environments are wide compared to the size of marsh features.

In the revised manuscript, we added the following text in section 4.2: 'As a proxy for the dissection of the platform by tidal creeks, we digitise tidal creek centrelines from the DEM. We then calculate the total length of tidal creeks included in the digitised platform divided by the platform surface area. We refer to this quantity as the Dissection Index (DI). In Fig. 11, we examine the capacity of the TIP-method to determine the area and perimeter of marsh platforms according to their dissection index. We find that for all test sites, TIP-detected area remains within 10% of the digitised area, whereas TIP-detected perimeter increases steadily with Dissection Index, confirming that the exclusion of tidal creeks by the TIP method is consistently stricter than by digitisation.'

Figure 11 was modified to have Dissection Index on the x-axis, thus highlighting the influence of dissection on the relative performances of the TIP method and digitisation.

**Comment 4**: *The paper could use some organizational finesse to improve the flow of the narrative. There are many instances where results are stated in the Methods section, and there is no Discussion section, but discussion elements are mixed in with Results. I would also consider adding a separate section for Validation following or within the Methods section to describe how you evaluated the performance of the TIP method. It seems very out of place in its current position (Results and Discussion). See specific comments below*

In order to make our manuscript clearer, we modified the structure to the following:

1. Introduction
2. Methodology
   (a) Test sites
   (b) Preprocessing Topographic Data
   (c) Scarp routing
   (d) Platform identification

***Comment 5****: The Results section is a bit messy. Perhaps consider organizing into a more logical manner. For instance, I like the idea of presenting results as a function of tidal range (or drainage density see comment 3). . .start with S1, describe, then go on to S2. . .and so on. Then, in a separate section (see comment 4) you could demonstrate the effects of using the filter on TIP results. I think this approach would be fine, because you already told us that you dont want to use the filter. . .and thats OK.*

As mentioned in our response to comment 4, we reorganised our manuscript to improve its clarity and better follow the order of the figures. We have followed some of the reviewers suggestions, for example more granularity in the results section as well as separating results and discussion. However, we we did not order the results on a site-by-site basis as we believe it would lead to a much longer results section, as well as making it more difficult to link results from different sites when illustrating our discussion and demonstrating the overall performance of the TIP method.

*P1L3: The productivity and even survival of salt marsh. . .(remove even).*
This modification was made.
*P1L3: . . .of salt marsh vegetation. . .(why vegetation here? Why not landscape? Seems out of place.)*
This was changed to 'the sustained existence of the salt marsh ecosystem'
*P1L5, P1L7, P1L15*
The suggested changes were made
*P1L20: . . ., it also suggests. . . (what is it?)*
This was changed to 'we suggest'
*P2L9: awkward, consider revising. . .perhaps something like, . . .makes monitoring the evolution of salt marshes imperative for management strategies and scientific endeavors. . ..*
The original text was replaced by: "makes monitoring the evolution of salt marshes crucial for developing management strategies that maintain the health of these ecosystems."
*P2L34: Right. . .but marsh platform slopes are on the order of 30cm total relief. . .and the overall structure is often misrepresented by lidar sensors with a nominal accuracy of +/- 15cm.*
The vertical accuracy (z-accuracy) of airborne lidar and photogrammetry may indeed be close to the size of the smallest scarps, which may be 30 cm or less in height for micro-tidal areas, immature platforms or marshes situated high in the tidal frame. If we consider an unvegetated surface (or a 'cleaned' DSM), we argue that the nominal accuracy is a combined product of georeferencing and distance-measurement accuracy of the lidar itself, the georeferencing generally accounting for the main part of the vertical error. The TIP method is focused on relative elevations in local neighbourhoods, and is therefore not very sensitive to the z-accuracy. If we consider a vegetated surface however, DEM processing (such as ground-return filtering and rasterisation) may indeed lead to higher and more locally disparate z-accuracy values on the platform than on the tidal flat. We argue in section 4.1 that the presence of vegetation induces positive errors, which plays in favour of the TIP method, as this artificially increases the platform height and therefore the scarp slope. To highlight these points, we have included an

example of very low relief marsh, Wax Lake Delta, and a marsh with low local relief, Morro Bay marsh, for which the TIP method succesfully identifies the marsh platform.

*P3L19: . . .horizontal resolutions. . .(do you mean horizontal extents? These are two very different things.)*

By horizontal resolution we mean the grid cell size of a rasterised DEM. The text was amended to read 'at varying grid cell sizes' to signify that all six sites were examined at different grid cell sizes.

*P3L20: remove dash after short.*

We made this change.

*P3L29: can you provide any technical specs for the lidar survey? Seasonality? Tides? Etc. . .*

We included a link to the metadata for Environment Agency lidar. The exact flight times relative to tides are unknown, however lidar surveys by the EA are conducted around low tide.

*P4L6: stronger than what?*

We now say: 'subject to a spring tidal range of 3.8 m and fluvio-tidal currents due to their estuarine fringing position'. *P4L9: . . .provides. . .(change to provide)*

We made this change.

*P4L10: what do you mean by numerous? How many more channels are at this site compared to the others? Consider using more physical descriptors throughout your study site description. What are the respective areas?*

Figure 11 of the revised manuscript provides a measure of the dissection of each marsh platform.

*P4L12: What do you mean by levels? Elevation? Water level?*

We replaced this with 'elevations'.

*P4L21: Why three times the horizontal resolution of the DEM? Why not 5 or 6?*

We now say: 'selected because it is the minimum radius needed to calculate slope with this method'.

*P4L31-33: At what scale is this problematic? 100s of kilometers? 10s of kilometers? I thought we were focused on relatively small areas of marsh landscape. . .what are the relative sizes of each study site (see also comment P4L10 above).*

The sites considered here are indeed small section of marshes. However, the local definition of kernels is unaffected by the DEM extent. The calculation time would however increase for larger marshes. At the time of writing, we have not tested the method on marshes larger than 12 km$^2$, for which the method did not encounter difficulties, despite the longer run times. Dimensions of the sites are included in the caption of Figure 10 of the revised manuscript. We replaced the original formulation by 'Likewise, although marsh platforms are locally higher than tidal flats and channels, this may not be the case for complex depositional environments (e.g. marshes sheltered by a sand spit), where long-shore declivity may cause portions of the tidal flats to be higher than distant emergent platforms' to better make our point.

*P5L9-22: Can you briefly describe what each of these means physically and the importance of information provided by each?*

Although the non-dimensional values of elevation and slope indeed have physical meaning, we did not wish to detail extensively as this would require investigation into formative processes for each site and considerably lengthen the manuscript. The aim of the manuscript is not to explore the history of the six test sites but rather to demonstrate the general applicability of the method, and these metrics are solely a means for us to apply the method over different landscapes.

*P5L24 (and throughout): be careful to avoid stating results in Methods section.*

We have now separated these as requested by the reviewer.

*P5L25: what is pdf? define.*

'pdf' is here defined as a probability distribution function: this is now clear in the text.

*P6L20: large number of true scarps? Or do you mean large number of misidentified scarps that are actually creek banks?*

This procedure produces a large number of scarps that could be creek banks and local DEM irregularities.

We now say 'This procedure produces a large number of potentially misidentified scarps, as small creeks within the platform and in higher portions of the tidal flat tend to be selected during this procedure.'

*P7L7: why 11?*

11 is a value we chose to obtain a significantly wider kernel. Other values have not been tested.

*P7L16 (and throughout), P7L27 (and throughout), P8L2*

We made these changes.

*P8L11 (throughout): Methods presented in results section. Consider providing a separate subsection in Methods for validation and then share results in the proper Results section.*

We did this: see response to Specific Comment 4.

*P8L12-15: Im guessing TP, FN, etc. . . are obtained from subtracting? Maybe show that in Methods.*

TP are obtained by counting the number of boolean True values for the detected marsh and the digitised marsh. Same for FP, etc. We now include the specifics in section 2.5 on performance metrics.

*P9L1: . . .the manual digitization. . .did you even discuss that in your Methods section? What software was used? Scale?*

The details were already provided in the Methods section.

*P9L14 - line number no longer applies in the revised manuscript: describe one figure at a time, and in chronological order.*

See response to Specific Comment 4.

*P9L19-29 - line number no longer applies in the revised manuscript: discussion in Results section. Consider revising.*

See response to Specific Comment 4.

*P10L23: Isnt this simply a transition zone between marsh platform and tidal flat?*

Although these zones correspond topographically to transition zones, they might not be vegetated (which was observed on aerial imagery for this site), and potentially unstable. We have therefore not called them 'pioneer zones' to avoid confusion with vegetated transition zones.

*P10L26: . . .saltings. . .why are you defining this here? You referred to salting earlier with no definition. Define earlier.*

Saltings were replaced by 'fallen blocks' to avoid confusion.

*P10L17: . . .yes, but its limited to erosional landscapes with obvious scarps.*

See response to comment 1. The reviewer quite correctly notes that this is limited to marsh landscapes with erosional scarps. This fits with our definition of 'salt marsh platform'. However this morphology is extremely common in salt marshes across a wide range of environments: many Atlantic and Gulf coast marshes in North America, many UK marshes (and all we have examined for this paper), marshes along the north coast of France, marshes in Italy, and these are just the examples that the authors have personally visited. So we believe the method is applicable to a significant fraction of global salt marshes.

*P11L9: . . .algae. . .why is this here?? Did you test for this or are you speculating? Maybe you could instead say that your method works independent of such environmental factors. . .its implied, but not exactly tested for in this paper.*

We see that this was inelegantly introduced into the paper. It is here for a reason, however: initially when we started identifying marshes it was suggested to us that we simply use optical techniques. However in our imagery, and in particular at Shell Bay, algae and biofilms are widespread making much of the landscape green. We wanted to make the point that even if the landscape is green from algae, or if there are widespread biofilms, we could still detect a marsh platform. Of course, large accumulations of macro-algae (kelp, etc.) might trick the method. We now clarify this by saying 'Furthermore, the presence of algae, kelp or duckweed as well as varying vegetation reflectance properties, which may induce specific calibrations with spectral methods (Morris et al., 2005), do not affect our results (barring mounds of stranded algae large enough to affect topography).'

*Figure 3, 5, 7, 9, 11*

The suggested changes were made.

*Figure 8: maybe a table would be a nice complement to this figure?*

A table for each subplot was added in Appendix A.

*Figure 10: scale?*

Scale is included in text to avoid clutter on these already crowded maps.

**Reviewer 2** We would like to thank the reviewer for their comments and positive response to our manuscript. In our revised manuscript we will make the following changes in light of the reviewer's helpful suggestions.

*My only concern is that the methodology is presented as a general tool for salt marsh and tidal flat identification, while I believe that its application is limited to the specific type of marshes presented in this study. I suggest the authors to:*

**Suggestion 1**: *better clarify the specs of the methodology that are tightly linked to the morphological characteristics of the specific study sites in order to make aware the user of the limits in applying the methodology*

We took the point of this comment, similar to comments made by Reviewer 1, and added text on the limitations of the method (especially in low relief or emergent marshes). We also added some more example marshes in Appendix B to demonstrate the method's results beyond the UK. See responses to reviewer 1 for specific text additions.

Again, we do feel it necessary to clarify that the TIP method is not site specific. It has been designed to apply to a wide variety of marshes and not only the test sites. We tested the method across a number of sites in the United Kingdom because these sites have a wide range of tidal range and wave climates. The basic features the method extracts, namely flat areas separated by scarps, are common to many marsh environments (albeit with some exceptions, which we will describe in the revised text). We acknowledge that the UK sites do not have a large variety of vegetation types, but in general the basic geometry of marshes is common across salt marshes, from the macrotidal Mont Saint Michel estuary to the microtidal Venice lagoon, which have rather different vegetation assemblages. Marshes along the Gulf and Atlantic coasts of North America also share the common morphology of flat areas separated by scarps. The stabilisation of deposited sediment and increased deposition rates (by direct trapping and velocity reduction) induced by vegetation are processes that occur in salt marshes regardless of their geographical location and local forcings. This process leads to the bifurcation of salt marsh platforms from tidal flats, leading to the formation of scarps (Mariotti and Fagherazzi 2010, DOI: 10.1029/2009JF001326). The TIP method therefore hinges only on the existence of a scarp and its on its representation on a DEM. The method does not depend on other morphological features save the absence of a very large river channel in the tidal flat.

In the revision we have explored marshes that are likely to be at the limits of the TIP method's ability to detect marshes and summarised for the reader the conditions that limit the method's accuracy, as suggested by the reviewer. We tested our method on the Wax Lake Delta, LA, a marsh with very low relief, and the Plum Island marsh, MA, a site heavily influenced by human activity, as well as Morro Bay, CA, where relief is locally very low. We find that the method can detect the marsh platform successfully in all three environments, but is challenged by the presence of prograding patches of vegetation.

**Suggestion 2**: *describe in more details the 6 study sites considered in this research underlying the specific peculiar morphological characteristics. This will allow the user/reader to decide if the*

*methodology may be applied to a different study site. Moreover, the authors refer to a 20 cm value to be subtracted to define the minimum local elevation for a platform pixel (pag. 7 lines 7-8). Also in this case a more precise explanation should be included so that the reader can judge if this is a value typical of the considered study sites or can be generalized.*

We have added more text on limitation of the method in the method and discussion sections, and have added an appendix with more sites with different vegetation assemblages and tidal regimes. For the 20 cm threshold we now say in the method section: 'The algorithm will not identify as separate platforms separated by scarps less than this elevation threshold, so on microtial marshes this threshold can be lowered. We address this limitation in the discussion and appendix. The threshold is necessary to prevent the algorithm from excluding pools and slight depressions in the platform surface.'

As mentioned in the methods section, the discussion (section 4.2) now reads: 'The morphological characteristics of prograding marshes are different from those of established platforms: consequently, vegetation patches and pioneer zones are not the object of the TIP method. Specifically, prograding margins and vegetation patches tend to have a relief and slope that are close to those of the tidal flat, making their outlines invisible to the scarp routing process. The combined absence of scarps and low relief of prograding marshes then interfere with the 20 cm leeway included in the platform filling process and cause an excess of false positives. Users may reduce this leeway to improve accuracy (see Fig.B2b1), but we discourage the use of the TIP method to identify vegetation patches and prograding margins.'

*Pag. 5 lines 20-25: in the text I do not see a description of the gray area in Fig. 3a.*
The grey area in Fig3a corresponds to the values of $P*$ to be excluded from the initial search space. This was already present in the caption.
*Pag 7 lines 7-10: Is the value 20cm applied to all the study sites? Could you please better explain how this specific value has been selected? is there a relation with the tidal excursion for example? Is this value specific for the English study sites?*
See comments above in response to suggestion 2. To summarise: the 20cm threshold works in all but one of our test cases including the microtidal site at Morro Bay. It fails at Wax Lake Delta where the total elevation range, including vegetation, is 80 cm. So the 20 cm threshold works in all but the most low relief sites. We have added text to this effect in the revision (specifics can be found in response to suggestion 2).
*Figure 12: the faded lines are difficult to see*
In the revision we slightly modified the colours of the lines. Faded lines are however covered by bright lines in most cases, and this was clarified in the caption.

[revised manuscript text omitted]